# DEGAP: Dynamic Entropy-Guided Attention Perturbation for Contrastive Decoding in Large Vision-Language Models

**Hyein Seo**[*1]  **Yuna Jeong**[*2]  **Mingyu Kang**[*2]  **Junhyeong Park**[*2]  **Yong Suk Choi**[1]

## Abstract

Large Vision–Language Models (LVLMs) have shown outstanding performance across various multimodal tasks, but they still suffer from hallucinations, where they generate incorrect information by relying on language priors without visual grounding. To alleviate this issue, prior work has explored contrastive decoding approaches that compare the output of an original LVLM with that of a contrast branch. However, existing methods typically generate contrast logits through preprocessing of the input image. Such input-level perturbations fail to sufficiently reflect the model's internal degree of visual reliance during the decoding process. To address this limitation, we propose Dynamic Entropy-Guided Attention Perturbation (DEGAP) for contrastive decoding in LVLMs. DEGAP performs contrastive decoding by directly perturbing visual attention and leveraging the resulting logits, without requiring any additional image preprocessing. To this end, we analyze the layer-wise effects of visual attention perturbations and, based on these observations, dynamically select the layers at which attention perturbation is applied according to the model's confidence. Experimental results on seven benchmarks demonstrate that DEGAP effectively mitigates various types of hallucinations and consistently outperforms state-of-the-art methods in general VQA performance.

## 1. Introduction

Large Vision-Language Models (LVLMs) have demonstrated strong performance across a wide range of multi-

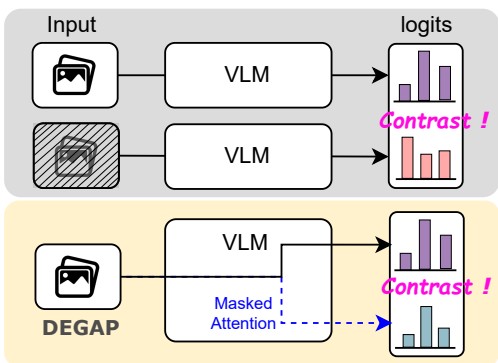

*Figure 1.* Comparison between prior contrastive decoding and DEGAP. (Top) Existing methods obtain contrast logits by externally perturbing the input image and running an additional VLM forward pass for the contrast branch. (Bottom) DEGAP constructs the contrast branch by masking visual attention through partial inference on subsequent layers.

modal tasks, including image captioning and visual question answering (Liu et al., 2023b; Li et al., 2023a; Ye et al., 2024). LVLMs integrate the generative and reasoning capabilities of Large Language Models (LLMs) with visual inputs. This enables them to serve as a strong foundation for multimodal systems that require open-ended understanding and instruction-following across diverse vision-language tasks. However, hallucination remains a persistent issue, where models generate plausible outputs even without visual grounding, undermining their reliability in practical use.

Hallucination in LVLMs refers to the generation of fluent and semantically coherent outputs that are not grounded in the input image. In particular, when a model relies excessively on learned language priors, it may generate plausible yet incorrect predictions that do not sufficiently reflect the given visual evidence (Lee et al., 2023; Yin et al., 2024; Liu et al., 2023a; 2024a). To mitigate this problem, instruction tuning (Liu et al., 2023a) and reinforcement learning (Sun et al., 2023; Gunjal et al., 2024) have been explored, but such approaches often require considerable computational cost and training time.

Recently, training-free contrastive decoding, which con-

*Equal contribution [1]Department of Computer Science, University of Hanyang, Seoul, Republic of Korea [2]Department of Artificial Intelligence, University of Hanyang, Seoul, Republic of Korea. Correspondence to: Yong Suk Choi <cys@hanyang.ac.kr>.

*Proceedings of the 43rd International Conference on Machine Learning*, Seoul, South Korea. PMLR 306, 2026. Copyright 2026 by the author(s).

*Figure 2.* Overview of our proposed DEGAP, which performs entropy-guided dynamic layer selection to determine the masking start layer $l_t^*$ at each decoding step. Our method then applies visual attention masking from $l_t^*$ onward to construct a contrast branch and performs contrastive decoding by combining the original and perturbed logits with efficient partial inference.

trasts token predictions between the original input and a perturbed input, has attracted attention. As an inference-time intervention, contrastive decoding provides a lightweight and broadly applicable way to suppress hallucination-prone predictions without additional training. Many existing methods obtain contrastive logits by externally perturbing the input, such as through input-level image modifications or additional modules (Leng et al., 2024; Favero et al., 2024; Wang et al., 2024; Rombach et al., 2022). However, such perturbations fail to sufficiently reflect the model's internal degree of reliance on visual information, limiting their ability to capture hallucinations arising from over-reliance on language priors. Furthermore, in many cases, obtaining contrastive logits requires an additional inference, increasing time cost. This overhead becomes even more pronounced when external generative models such as diffusion models are used (Zhang et al., 2025; Chen et al., 2024).

In this work, we analyze how different perturbations affect the layer-wise output distributions of LVLMs, motivated by the observation that visual information is utilized differently across layers. Building on this analysis, we propose DEGAP, a contrastive decoding method that strengthens visual grounding when LVLMs generate responses without additional training or external input perturbations. Specifically, we perform entropy-guided dynamic layer selection to determine the intervention point conditioned on the input and each decoding token, and apply visual attention masking to the layers after the selected point. By constructing contrastive logits that reflect the model's hallucination tendencies, our method can effectively reduce them during generation. Moreover, DEGAP efficiently obtains contrastive logits by reusing the hidden state before masking and performing partial inference on subsequent layers, avoiding an extra full forward pass for the contrastive branch, as shown

in Figure 1. Extensive experiments across seven benchmarks demonstrate that DEGAP consistently outperforms state-of-the-art methods in both hallucination mitigation and general multimodal performance. Our main contributions are summarized as follows:

- We propose DEGAP, a contrastive decoding method that induces contrastive logits by masking visual attention, without additional training or external input perturbations.

- We analyze the impact of perturbations on VLMs and the correlation with entropy, and introduce entropy-guided dynamic layer selection to dynamically determine the optimal intervention point.

- Extensive experiments demonstrate that DEGAP consistently outperforms existing methods, mitigating hallucinations, with lower additional latency than prior contrastive decoding methods.

## 2. Related Work

### 2.1. Hallucination in LVLMs

Autoregressive LLMs have advanced rapidly, leading to the sequential emergence of increasingly capable language models (Touvron et al., 2023; Chowdhery et al., 2023; Chiang et al., 2023). Based on these advancements, LVLMs have been developed by extending the capabilities of LLMs to incorporate visual information. These models demonstrate strong performance across various multimodal tasks (Liu et al., 2023b; Dai et al., 2023; Ye et al., 2024; Sun et al., 2023).

Despite these successes, LVLMs frequently suffer from

hallucinations, generating content that is inconsistent with the given image (Li et al., 2023b; Liu et al., 2024a; Bai et al., 2024). Various methods have been proposed to mitigate hallucinations. For example, one approach involves training with large-scale synthesized multimodal preference data (Wu et al., 2025; Xie et al., 2024; Zhou et al., 2024). Another line of work focuses on learning post-hoc revisors to correct the generated output after initial generation (Zhou et al., 2023; Yin et al., 2024). However, these methods typically require substantial annotated data, extensive computational resources, or costly fine-tuning procedures, which limit their scalability and practical deployment.

In this work, we present a method that mitigates hallucinations and promotes visually grounded responses without requiring additional training or external data.

## 2.2. Decoding Based on Visual Grounding

To mitigate hallucinations caused by language priors in LVLMs, recent studies have proposed novel decoding methods that incorporate visual information more directly. In particular, contrastive decoding approaches (Leng et al., 2024; Favero et al., 2024) aim to improve response reliability by leveraging contrastive logits derived from distorted inputs. These logits are typically constructed through external image manipulations, such as adding Gaussian noise (Leng et al., 2024), modifying the field of view (Deng et al., 2024), applying transformations such as flipping, rotation, and cropping (Woo et al., 2024), or even generating entirely new images (Zhang et al., 2025). However, these methods depend on external models or modifications, which may degrade logit quality and introduce additional computational overhead.

DeCo (Wang et al., 2025) is a closely related decoding-time method that leverages layer-wise internal signals to mitigate hallucinations. It selects an intermediate layer containing reliable visual evidence and refines the final prediction through logit-level correction. While DEGAP also performs layer-wise analysis during decoding, its objective is fundamentally different. Rather than reinforcing reliable signals, our method identifies where visual information should be perturbed to construct an effective contrast branch. In this sense, DeCo is a correction-based approach, whereas DE-GAP directly suppresses hallucination-inducing signals.

Another decoding approach, DropoutDecoding (Fang et al., 2024), estimates the uncertainty of each visual token individually and drops the uncertain ones during decoding. However, this method relies on token-level uncertainty estimation and ensemble decoding, which substantially increase computational cost. Furthermore, it does not directly suppress hallucinations arising from language priors.

In contrast, our method induces and eliminates hallucinatory

patterns by masking visual attention after a specific layer, without modifying the input image. This design directly suppresses hallucinatory logits arising from language priors, resulting in more reliable and faithful responses. Moreover, unlike DropoutDecoding, DEGAP hierarchically analyzes the model's predictive confidence to identify the optimal point for suppressing visual information, thereby mitigating hallucinations more efficiently.

## 3. Method

### 3.1. Preliminary

**Generation of LVLMs.** LVLMs generate responses conditioned on both the input image and the textual query. Typically, the image is encoded into visual features using a vision encoder (e.g., ViT (Dosovitskiy et al., 2020)), and is projected into the language embedding space through a vision-language alignment module (e.g., Q-Former (Dai et al., 2023), linear projection (Liu et al., 2023b)). This process produces visual tokens $v$, which are concatenated with the tokenized query $\mathbf{x}$ and fed into an autoregressive language model. The model employs causal attention to generate response tokens $\mathbf{y} = [y_0, \ldots, y_T]$ one at a time. At each time step $t$, the model generates a token $y_t$, which represents the output at that step. This process can be formally described as:

$$y_t \sim p_\theta(y_t \mid v, \mathbf{x}, \mathbf{y}_{<t}) \propto \exp f_\theta(y_t \mid v, \mathbf{x}, \mathbf{y}_{<t}), \quad (1)$$

where $\mathbf{y}_{<t}$ denotes all tokens generated before time step $t$, and $f_\theta$ represents the logit distribution over the next token. The probability distribution is computed by mapping the final hidden state $h_t^L$ at the last layer $L$ to the vocabulary $\mathcal{V}$ using a linear projection matrix $W_\mathcal{V} \in \mathbb{R}^{|\mathcal{V}| \times d}$ as follows:

$$p_\theta(y_t \mid h_t^L) = \text{softmax}(W_\mathcal{V} h_t^L) \in \mathbb{R}^{|\mathcal{V}|} \quad (2)$$

**Logit Lens.** We use logit lens to probe how intermediate hidden states are mapped to token predictions and to track how this mapping evolves across layers under different conditions. Concretely, for each decoding step $t$ and layer $l$, we take the intermediate hidden state $h_t^l$ and project it to the vocabulary space using the model's output projection matrix $W_\mathcal{V}$. This yields a layer-wise token distribution:

$$p_\theta^{(l)}(y_t \mid h_t^l) = \text{softmax}(W_\mathcal{V} h_t^l) \in \mathbb{R}^{|V|} \quad (3)$$

The distribution $p_\theta^{(l)}$ can be interpreted as the model's "best guess" of the next token if decoding were to stop at layer $l$. By computing $p_\theta^{(l)}$ for all layers, we obtain a layer-by-layer view of how the next-token distribution forms during the forward pass.

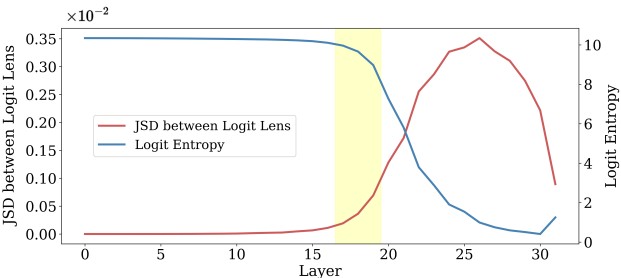

*Figure 3.* Layer-wise perturbation effects and logit entropy in LVLM outputs. Jensen–Shannon Divergence (JSD) between token distributions from the original and perturbed images measured by a logit lens at each layer, and the original branch logit lens entropy.

## 3.2. Experimental Setup for Analysis

For the analysis, we randomly sample 1,000 images from MSCOCO 2014 validation set (Lin et al., 2014). The model used for the analysis is LLaVA-1.5 7B (Liu et al., 2023b). We additionally report our analysis on other LVLMs, including InstructBLIP (Dai et al., 2023) and mPLUG-Owl2 (Ye et al., 2024), in Appendix B.1. For each image, the model is prompted with the following input to generate a response: *"Please describe this image in detail."*

## 3.3. Layer-wise Analysis of Attention Perturbation

As noted in prior studies (Leng et al., 2024; Favero et al., 2024), the effectiveness of contrastive decoding depends on how well the generated contrastive logits reflect hallucinations that arise when visual grounding is weakened. Motivated by the observation that LVLMs leverage visual information differently across layers (Jiang et al., 2025), we analyze the layer-wise effects of attention perturbations implemented by masking visual attention on the model's output distributions. Specifically, we set the attention score over visual tokens to $-\infty$, which yields zero attention weights after softmax. Using this perturbation, we employ a logit lens to project each layer's hidden representations into the vocabulary space. We then measure the divergence between the token distributions from the original and perturbed branches using Jensen–Shannon Divergence (JSD) (Menéndez et al., 1997), i.e., $\mathrm{JSD}(p_\theta^{(l)}(y_t \mid v, \mathbf{x}, \mathbf{y}_{<t}), p_\theta^{(l)}(y_t \mid v', \mathbf{x}, \mathbf{y}_{<t}))$, where $v'$ denotes the perturbed input with visual attention masked. As shown in Figure 3, the divergence between the original and contrastive logits remains negligible in the early layers, but increases sharply in the later layers. This shows that the perturbation effects are not uniformly distributed across the network but are concentrated at deeper layers.

Interestingly, we observe that this pattern is closely aligned with the evolution of the output entropy of the model. To investigate this relationship, we explicitly measure the layer-wise logit entropy $H_{l,t}$ at each layer using the logit lens,

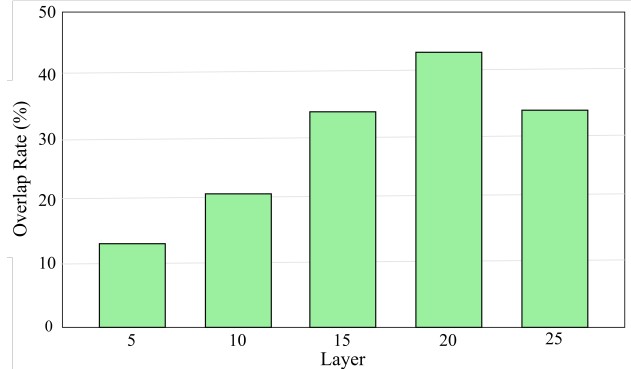

*Figure 4.* Overlap rate between hallucinated objects from the original branch and the contrast branch induced by layer-wise visual attention masking, averaged over 1,000 sampled instances.

defined as

$$H_{l,t} = -\sum_{w \in \mathcal{V}} p_\theta^{(l)}(y_t = w \mid h_t^l) \log p_\theta^{(l)}(y_t = w \mid h_t^l) \quad (4)$$

As shown in Figure 3, the layer-wise logit entropy analysis of the original branch reveals that the layers at which perturbation effects become prominent coincide with the point where entropy begins to decrease. This alignment indicates that the divergence between the original and contrastive logits arises as the output distribution becomes more concentrated and prediction confidence starts to form. Consequently, perturbations applied at this stage have a more direct and pronounced effect on the output distribution.

Based on this insight, we adopt visual attention masking as an internal intervention mechanism for controlled layer-wise perturbation. We then analyze its effects by applying visual attention masking from different starting layers and measuring the overlap between the resulting contrastive logits and MSCOCO hallucinated objects generated by the original branch. As shown in Figure 4, masking visual attention from earlier layers results in only marginal overlap, whereas masking visual attention starting from middle-to-late layers leads to a substantially higher overlap with hallucinated objects. These results suggest that the model's hallucination tendencies are more clearly exposed when visual information is removed at stages where the output distribution has already begun to consolidate. Additional details are provided in Appendix B.1.

## 3.4. Dynamic Entropy-Guided Layer Selection

We propose DEGAP, a decoding strategy that constructs contrastive logits by masking visual attention weights inside the decoder of LVLMs. The overall architecture is illustrated in Figure 2. The key idea is to (i) decide when to start visual attention masking dynamically for each input and decoding step, and (ii) reduce inference overhead via partial inference by reusing the output of the layer immediately

before masking from the original branch, rather than running a full second forward pass.

As shown in the analysis in Section 3.3, attention perturbations do not affect all layers uniformly and begin to have an effect from the middle to later layers. This phenomenon is related to the region where the layer-wise logit entropy starts to decrease sharply. However, using a fixed masking layer from the middle or later layers is brittle, as the optimal intervention point varies across models, datasets, and inputs. To avoid additional tuning, we dynamically select the masking start layer at each decoding step using the entropy curve across layers.

In practice, logit entropy $H_{l,t}$ often stays nearly flat in early layers, then exhibits a sharp drop at an intermediate depth, and may subsequently plateau again or even slightly increase near the final layers. To automatically identify, for each input and decoding step, the layer at which this sharp decrease begins, we define the masking start layer $l_t^*$ as the descending knee point of the entropy curve (Satopaa et al., 2011), i.e., the point where the curve shifts from a flat regime to a steep downward regime.

To avoid being affected by potential plateauing or re-increase near the end of the network, we detect this knee only up to the layer where the entropy attains its minimum for token $t$. Specifically, let

$$l_{\min} = \underset{l \in \{0,\dots,L-1\}}{\arg\min} H_{l,t} \quad (5)$$

We compute the knee by fitting a two-segment piecewise-linear model (Muggeo, 2003) to the truncated layer-wise entropy sequence $\{(l, H_{l,t})\}_{l=0}^{l_{\min}}$ with a candidate split $k$. For each $k \in \{1, \dots, l_{\min} - 2\}$, we fit two straight lines via least squares to the two intervals: a first line $\hat{a}_1(k)l + \hat{b}_1(k)$ over $[0, k]$ and a second line $\hat{a}_2(k)l + \hat{b}_2(k)$ over $[k+1, l_{\min}]$. Each segment is modeled in the form $y = al + b$, where $a$ denotes the slope and $b$ denotes the $y$-intercept.

$$E_t(k) = \frac{1}{k+1} \sum_{l=0}^{k} \left( H_{l,t} - \left(\hat{a}_1(k)l + \hat{b}_1(k)\right) \right)^2$$
$$+ \frac{1}{l_{\min} - k} \sum_{l=k+1}^{l_{\min}} \left( H_{l,t} - \left(\hat{a}_2(k)l + \hat{b}_2(k)\right) \right)^2 \quad (6)$$

We then define the masking start layer $l_t^*$ as

$$l_t^* = \underset{k \in \{1, \dots, l_{\min} - 2\}}{\arg\min} E_t(k) \quad (7)$$

This rule does not rely on any absolute entropy threshold and thus adapts automatically to different entropy scales across models, datasets, and inputs.

Given $l_t^*$, we construct a contrast branch by fully blocking access to visual evidence from layer $l_t^*$ onward (100% mask-ing of visual attention) through the final decoder layer. This can be summarized as:

$$\mathbf{h}_t^{l+1} = \begin{cases} \text{Layer}_l(\mathbf{h}_t^l), & l < l_t^*; \\ \text{Layer}_l^{\text{masked}}(\mathbf{h}_t^l), & l \geq l_t^*, \end{cases} \quad (8)$$

where $\text{Layer}_l^{\text{masked}}(\cdot)$ denotes a decoder layer with visual attention fully masked.

Crucially, we obtain the contrast branch final logits without performing a full second forward pass. At decoding step $t$, we first run the original (unmasked) branch as usual and cache the hidden state right before the intervention point, i.e., the output of layer $(l_t^* - 1)$. Starting from this cached state, we then execute only the remaining layers $l_t^*, \dots, L-1$ with visual attention fully disabled, producing the contrast branch logits with a partial forward computation. As a result, the additional cost is limited to re-running only the later layers, reducing latency compared to contrastive decoding methods that require an additional full inference run, as shown in Section 4.3.

Let $f_\theta(y_t \mid v, \mathbf{x}, \mathbf{y}_{<t})$ denote the logits from the original branch, and let $f_\theta^*(y_t \mid v, \mathbf{x}, \mathbf{y}_{<t})$ denote the logits from the attention-masked branch produced by our visual masking strategy. Using these two logits, we perform contrastive decoding by combining them as

$$p_\theta(y_t) = \text{Softmax}\Big[ (1 + \alpha) f_\theta(y_t \mid v, \mathbf{x}, \mathbf{y}_{<t})$$
$$- \alpha f_\theta^*(y_t \mid v, \mathbf{x}, \mathbf{y}_{<t}) \Big], \quad (9)$$

where $\alpha$ controls the strength of the contrastive term. Because the attention-masked branch has reduced access to visual evidence, its logits are more driven by the language prior; subtracting them therefore penalizes tokens that are favored without visual grounding, helping mitigate hallucination.

## 4. Experiments

### 4.1. Experimental Settings

**Models**  We integrate our method with three LVLMs: LLaVA-1.5, InstructBLIP, and mPLUG-Owl2. All models are configured with 7B parameters. LLaVA-1.5 and InstructBLIP utilize Vicuna 7B as the language decoder, while mPLUG-Owl2 is built upon the LLaMA 7B backbone.

**Benchmarks**  We evaluate DEGAP on seven benchmarks covering both hallucination mitigation and general multimodal reasoning. For hallucination mitigation, we use CHAIR (Rohrbach et al., 2018), POPE (Li et al., 2023b), MME-Hallucination (Fu et al., 2023), and HallusionBench (Guan et al., 2024). For general tasks, we evaluate on MM-Bench (Liu et al., 2024b), LLaVA-Bench (Liu et al., 2023b),

*Table 1.* **Results on the hallucination subsets of MME, HallusionBench, MMBench, and MM-Vet.** The underlying LVLM is LLaVA-1.5. For each subset, we report the mean MME scores and standard deviations across three repeated runs.

| Method | Object-level | | Attribute-level | | MME Score | Hallusion Bench | MMBench | MM-Vet |
|---|---|---|---|---|---|---|---|---|
| | Existence ↑ | Count ↑ | Position ↑ | Color ↑ | | | | |
| Regular | 183.33(±2.89) | 125.00(±10.41) | 121.67(±5.00) | 143.33(±6.01) | 573.33(±17.64) | 32.7 | 40.0 | 22.9 |
| VCD | 183.33(±7.64) | 136.66(±7.64) | 122.78(±3.85) | 143.33(±6.67) | 586.10(±9.76) | 35.8 | 57.0 | 25.5 |
| M3ID | 186.67(±2.89) | 133.89(±6.94) | 124.44(±0.96) | 148.34(±2.89) | 593.34(±8.67) | 35.5 | 54.7 | 26.7 |
| RITUAL | 186.67(±2.89) | 138.89(±12.28) | 111.67(±5.00) | 160.00(±5.00) | 597.23(±17.97) | 34.9 | 57.3 | 27.4 |
| ICD | 185.00(±5.00) | 127.78(±10.05) | 122.78(±3.47) | 159.45(±2.55) | 595.01(±6.67) | 36.7 | 45.1 | 24.7 |
| OPERA | 183.33(±6.45) | 137.22(±6.31) | 122.78(±2.55) | 155.00(±5.00) | 598.33(±10.41) | 36.6 | 56.7 | 25.8 |
| HALC | 183.33(±0.00) | 133.33(±5.77) | 107.92(±3.69) | 155.00(±5.00) | 579.58(±9.07) | 35.6 | 53.0 | 26.2 |
| DROPOUT | 185.00(±8.86) | 131.11(±4.82) | 122.44(±5.36) | 149.45(±2.55) | 590.00(±3.34) | 36.9 | 59.8 | 27.6 |
| DeGF | 180.00(±0.00) | 136.66(±16.07) | 122.22(±4.20) | 166.66(±2.89) | 605.54(±15.48) | 37.4 | 60.4 | 27.1 |
| **DEGAP** | **193.33**(±2.89) | **145.00**(±7.64) | **125.00**(±2.89) | **167.78**(±2.55) | **631.11**(±4.19) | **37.8** | **62.4** | **29.4** |

*Table 2.* **Results on CHAIR** with $C_S$ (CHAIRs), $C_I$ (CHAIRi), Rec. (Recall), Len. (average response length in tokens). The number of max new tokens is set to 64. Best results are **bold** and second-best results are underlined.

| Method | LLaVA-1.5 | | | | InstructBLIP | | | |
|---|---|---|---|---|---|---|---|---|
| | $C_S$ ↓ | $C_I$ ↓ | Rec. ↑ | Len. ↑ | $C_S$ ↓ | $C_I$ ↓ | Rec. ↑ | Len. ↑ |
| Regular | 27.2 | 9.8 | 57.0 | 53.5 | 27.0 | 9.1 | 59.5 | 54.0 |
| VCD | 26.6 | 9.4 | 62.9 | 53.7 | 27.4 | 9.3 | 63.4 | 54.3 |
| M3ID | 27.6 | 8.5 | 64.1 | 53.2 | 29.8 | 10.4 | 62.2 | 53.5 |
| RITUAL | 21.4 | 7.1 | 61.6 | 54.8 | 25.8 | 8.9 | 65.2 | 55.7 |
| ICD | 23.2 | 7.7 | 63.8 | 54.3 | 29.8 | 10.2 | 61.6 | 54.6 |
| OPERA | 21.8 | 7.1 | 60.6 | 52.8 | 23.4 | 8.6 | 62.7 | 53.5 |
| HALC | 21.2 | 6.9 | 58.3 | 53.5 | 29.6 | 11.3 | 61.8 | 55.1 |
| DROPOUT | 22.2 | 8.4 | 58.6 | 53.2 | 25.6 | 10.9 | 55.3 | 50.6 |
| DeGF | 22.4 | 7.6 | 61.0 | 54.2 | 23.8 | 8.2 | 61.9 | 54.9 |
| **DEGAP** | **19.1** | **6.3** | **65.2** | 54.5 | **22.1** | **7.7** | 65.1 | 55.3 |

and MM-Vet (Yu et al., 2023). Detailed descriptions of these benchmarks are provided in Appendix A.1.

**Baselines** We compare our method against state-of-the-art hallucination mitigation methods. These include four representative contrastive decoding methods: VCD (Leng et al., 2024), M3ID (Favero et al., 2024), RITUAL (Woo et al., 2024), and ICD (Wang et al., 2024). For CHAIR and MME benchmarks, we additionally include DeGF (Zhang et al., 2025), OPERA (Huang et al., 2024), HALC (Chen et al., 2024), and DROPOUT (Fang et al., 2024) as other decoding methods. A detailed description of each baseline is provided in Appendix A.3.

**Implementation Details** Following VCD (Leng et al., 2024), we set $\alpha = 1$ for all tasks. All experiments are conducted on an NVIDIA RTX A5000 GPU. For all benchmark evaluations, we report the average results across three repeated runs.

### 4.2. Main Results

**Results on Hallucination Mitigation Tasks** On CHAIR, Table 2 shows that DEGAP achieves the lowest CHAIRs and

CHAIRi while maintaining strong recall on both LLaVA-1.5 and InstructBLIP. Compared to strong baselines, DEGAP reduces $C_S$ and $C_I$ by 2.1 and 0.6 on LLaVA-1.5, and by 1.3 and 0.9 on InstructBLIP, with comparable response length. Moreover, DEGAP improves recall by 8.2 on LLaVA-1.5 and by 5.6 on InstructBLIP, indicating reduced hallucinations without sacrificing relevance.

On MME and HallusionBench (Table 1), DEGAP attains the best overall scores and consistently improves all sub-tasks. Notably, it boosts Count and Position by +20.00 and +3.33 over the regular model, demonstrating robustness even in complex scenes.

On POPE (Table 3), DEGAP consistently improves over regular decoding across Random/Popular/Adversarial splits and multiple backbones, with gains up to +7.09 in accuracy and +6.73 in F1. Averaged over datasets, accuracy increases by +1.50 for LLaVA-1.5, +3.51 for InstructBLIP, and +2.69 for mPLUG-Owl2. While ICD slightly outperforms DE-GAP in a few Popular/Adversarial cases on LLaVA-1.5, DEGAP provides more consistent gains across backbones while keeping precision comparable and improving recall on average, yielding a strong Precision–Recall trade-off.

**Results on General Tasks** We further assess DEGAP on general-purpose benchmarks that measure overall multimodal capability, including MMBench, MM-Vet, and LLaVA-Bench. As summarized in Table 1, DEGAP achieves the best performance on both MMBench and MM-Vet, surpassing the strongest baseline by +2.0 and +1.8, respectively, indicating a balanced and generalizable improvement beyond hallucination-specific settings. The full MM-Bench and MM-Vet results are provided in Appendix B.2 and Appendix B.3.

LLaVA-Bench employs GPT-4o as an automated evaluator, rating both Accuracy (Acc.) and Detailedness (Det.) on a 1–10 scale. Table 5 reports the average pairwise scores

*Table 3.* **Results on the POPE.** Higher (↑) Accuracy, Precision, Recall, and F1 indicate better performance. "Regular" decoding denotes direct sampling.

| | Setup | Method | LLaVA-1.5 | | | | InstructBLIP | | | | mPLUG-Owl2 | | | |
|---|---|---|---|---|---|---|---|---|---|---|---|---|---|---|
| | | | Acc. ↑ | Prec. ↑ | Recall↑ | F1 ↑ | Acc. ↑ | Prec. ↑ | Recall↑ | F1 ↑ | Acc. ↑ | Prec. ↑ | Recall↑ | F1 ↑ |
| MS-COCO | Random | Regular | 85.22 | 84.79 | 86.93 | 85.85 | 82.60 | 82.17 | 83.27 | 82.72 | 80.65 | 77.87 | 87.27 | 82.30 |
| | | VCD | 85.02 | 84.10 | 87.47 | 85.75 | 84.90 | 85.30 | 84.33 | 84.81 | 82.54 | 81.12 | 86.20 | 83.58 |
| | | M3ID | 86.47 | 85.34 | 88.07 | 86.68 | 84.40 | 85.77 | 83.60 | 84.67 | 82.51 | 79.90 | 88.27 | 83.88 |
| | | RITUAL | 88.56 | 89.45 | 88.20 | 88.82 | 87.83 | 88.90 | 86.47 | 87.60 | 84.78 | 82.85 | 88.87 | 85.75 |
| | | ICD | 86.77 | 87.69 | 86.47 | 87.08 | 85.43 | 86.55 | 84.93 | 85.73 | 82.16 | 79.39 | 88.33 | 83.62 |
| | | **DEGAP** | **89.30** | **89.60** | **88.91** | **89.25** | **89.69** | **91.55** | **87.45** | **89.45** | **85.78** | **83.65** | **88.95** | **86.22** |
| | Popular | Regular | 81.93 | 78.89 | 87.20 | 82.84 | 76.37 | 73.73 | 81.93 | 77.61 | 76.53 | 72.26 | 86.13 | 78.59 |
| | | VCD | 81.90 | 78.60 | 87.67 | 82.89 | 77.73 | 75.06 | 83.07 | 78.86 | 77.67 | 74.01 | 85.07 | 79.21 |
| | | M3ID | 83.17 | 80.24 | 88.00 | 83.94 | 78.33 | 75.30 | 84.33 | 79.56 | 77.23 | 72.86 | 86.80 | 79.22 |
| | | RITUAL | 85.30 | 83.53 | 87.93 | 85.68 | 82.13 | 79.35 | 86.87 | 82.94 | 80.07 | 75.57 | 88.87 | 81.68 |
| | | ICD | 83.67 | 82.04 | 86.20 | 84.07 | 78.40 | 75.45 | 84.20 | 79.58 | 78.23 | 73.54 | 88.20 | 80.21 |
| | | **DEGAP** | **85.81** | **83.92** | **88.60** | **86.20** | **82.95** | **79.96** | **87.94** | **83.76** | **81.23** | **77.17** | **88.70** | **82.53** |
| | Adversarial | Regular | 76.93 | 72.35 | 87.20 | 79.08 | 73.93 | 70.35 | 82.73 | 76.04 | 73.20 | 68.43 | 86.13 | 76.27 |
| | | VCD | 76.47 | 71.62 | 87.67 | 78.84 | 76.97 | 73.70 | 83.87 | 78.45 | 74.60 | 70.34 | 85.07 | 77.01 |
| | | M3ID | 77.77 | 73.05 | 88.00 | 79.83 | 75.73 | 72.06 | 84.07 | 77.60 | 73.70 | 68.78 | 86.80 | 76.75 |
| | | RITUAL | 79.07 | 74.47 | 88.47 | 80.87 | 78.63 | 74.45 | 87.20 | 80.32 | 75.70 | 70.34 | 88.87 | 78.53 |
| | | ICD | 79.10 | **75.48** | 86.20 | 80.49 | 76.70 | 73.19 | 84.27 | 78.34 | 74.60 | 69.34 | 88.20 | 77.64 |
| | | **DEGAP** | **79.92** | 75.21 | **89.27** | **81.64** | **79.82** | **75.72** | **87.78** | **81.31** | **77.33** | **72.26** | 88.71 | **79.64** |
| A-OKVQA | Random | Regular | 81.50 | 75.64 | 92.93 | 83.40 | 81.37 | 77.89 | 87.60 | 82.46 | 76.63 | 71.19 | 89.47 | 79.29 |
| | | VCD | 82.27 | 76.42 | 93.33 | 84.03 | 81.83 | 78.47 | 87.73 | 82.85 | 78.30 | 73.39 | 88.80 | 80.36 |
| | | M3ID | 83.50 | 78.09 | 93.13 | 84.95 | 83.13 | 79.20 | 89.87 | 84.20 | 77.87 | 72.26 | 90.47 | 80.34 |
| | | RITUAL | 85.47 | 80.26 | 94.07 | 86.62 | 87.20 | 83.86 | 92.13 | 87.80 | 80.20 | 74.38 | 92.13 | 82.31 |
| | | ICD | 84.30 | 79.79 | 91.87 | 85.40 | 82.70 | 78.77 | 89.53 | 83.81 | 78.53 | 72.53 | 91.87 | 81.06 |
| | | **DEGAP** | **86.31** | **81.05** | **94.78** | **87.38** | **87.84** | **84.18** | **93.19** | **88.46** | **81.73** | **76.01** | **92.72** | **83.54** |
| | Popular | Regular | 75.33 | 68.74 | 92.93 | 79.02 | 74.67 | 69.54 | 87.8 | 77.61 | 71.33 | 65.64 | 89.53 | 75.75 |
| | | VCD | 75.67 | 68.97 | 93.33 | 79.32 | 75.40 | 70.44 | 87.53 | 78.06 | 72.87 | 67.34 | 88.80 | 76.60 |
| | | M3ID | 77.20 | 70.63 | 93.13 | 80.33 | 77.10 | 71.84 | 89.13 | 79.56 | 72.83 | 66.88 | 90.47 | 76.91 |
| | | RITUAL | 78.63 | 71.77 | 94.40 | 81.54 | 78.27 | 71.68 | 93.27 | 81.13 | 74.30 | 67.91 | 92.13 | 78.19 |
| | | ICD | 78.27 | **72.22** | 91.87 | 80.87 | 76.17 | 71.16 | 88.00 | 78.69 | 72.13 | 65.87 | 91.87 | 76.73 |
| | | **DEGAP** | **79.14** | 72.21 | **94.75** | **81.96** | **79.86** | **73.49** | **93.42** | **82.27** | **74.82** | **68.16** | **93.14** | **78.72** |
| | Adversarial | Regular | 68.00 | 62.21 | 91.73 | 74.14 | 69.33 | 64.02 | 88.27 | 74.22 | 66.00 | 60.76 | 90.33 | 72.65 |
| | | VCD | 67.57 | 61.61 | 93.20 | 74.18 | 70.83 | 65.56 | 87.80 | 75.06 | 66.70 | **61.52** | 89.20 | 72.82 |
| | | M3ID | 68.20 | 62.17 | 93.00 | 74.52 | 69.83 | 64.39 | 88.73 | 74.63 | 65.73 | 60.50 | 90.67 | 72.57 |
| | | RITUAL | 68.17 | 62.01 | 93.80 | 74.66 | 70.53 | 64.30 | 92.33 | 75.81 | 65.90 | 60.45 | 92.00 | 72.96 |
| | | ICD | **69.23** | **63.24** | 91.87 | 74.91 | 69.53 | 63.94 | 89.60 | 74.63 | **66.23** | 60.72 | 91.93 | 73.14 |
| | | **DEGAP** | 68.88 | 62.42 | **94.91** | **75.31** | **71.71** | 65.08 | **93.70** | **76.81** | 66.73 | 60.96 | **93.05** | **73.66** |
| GQA | Random | Regular | 81.87 | 75.87 | 93.47 | 83.75 | 79.87 | 77.02 | 85.13 | 80.87 | 77.13 | 72.39 | 87.73 | 79.32 |
| | | VCD | 81.43 | 75.12 | 94.00 | 83.51 | 81.50 | 78.97 | 85.87 | 82.27 | 80.50 | 77.15 | 86.67 | 81.63 |
| | | M3ID | 82.70 | 76.64 | 94.07 | 84.47 | 81.03 | 77.76 | 86.93 | 82.09 | 78.67 | 74.00 | 88.40 | 80.56 |
| | | RITUAL | 85.53 | 79.35 | 96.07 | 86.91 | 84.63 | 82.33 | 88.20 | 85.16 | 81.83 | 76.63 | 91.60 | 83.45 |
| | | ICD | 84.20 | 78.79 | 93.60 | 85.56 | 81.33 | 78.28 | 86.73 | 82.29 | 79.30 | 74.59 | 88.87 | 81.11 |
| | | **DEGAP** | **86.20** | **80.08** | **96.37** | **87.47** | **85.54** | 82.29 | **90.58** | **86.24** | **83.01** | **77.88** | **92.20** | **84.44** |
| | Popular | Regular | 72.73 | 66.07 | 93.47 | 77.42 | 72.33 | 67.59 | 85.80 | 75.62 | 70.10 | 64.82 | 87.93 | 74.63 |
| | | VCD | 71.30 | 64.65 | 94.00 | 76.61 | 73.63 | 69.07 | 85.60 | 76.45 | 71.33 | 66.33 | 86.67 | 75.14 |
| | | M3ID | 72.77 | 65.97 | 94.07 | 77.55 | 73.40 | 68.67 | 86.07 | 76.39 | 70.67 | 65.26 | 88.40 | 75.08 |
| | | RITUAL | 73.57 | 66.27 | 96.00 | 78.41 | 75.17 | 69.71 | 89.00 | 78.18 | 73.43 | 67.19 | 91.60 | 77.52 |
| | | ICD | 73.70 | **67.13** | 92.87 | 77.93 | 74.67 | 69.51 | 87.87 | 77.62 | 70.50 | 64.99 | 88.87 | 75.08 |
| | | **DEGAP** | **73.98** | 66.56 | **96.37** | **78.74** | **76.89** | **70.99** | **90.94** | **79.74** | **74.37** | **67.89** | **92.49** | **78.30** |
| | Adversarial | Regular | 68.23 | 62.22 | 92.87 | 74.51 | 69.50 | 64.92 | 84.87 | 73.56 | 67.77 | 62.39 | 89.47 | 73.51 |
| | | VCD | 68.07 | 61.89 | 94.07 | 74.66 | 70.37 | 65.55 | 85.87 | 74.34 | 69.80 | **65.02** | 85.73 | 73.95 |
| | | M3ID | 68.43 | 62.21 | 93.93 | 74.85 | 68.00 | 63.33 | 85.53 | 72.77 | 68.47 | 63.15 | 88.67 | 73.77 |
| | | RITUAL | 68.33 | 61.90 | 95.33 | 75.07 | 70.33 | 64.75 | 89.27 | 75.06 | 68.07 | 62.25 | 91.80 | 74.19 |
| | | ICD | **69.60** | **63.39** | 92.80 | 75.32 | 69.27 | 64.25 | 86.87 | 73.87 | 68.53 | 63.19 | 88.80 | 73.84 |
| | | **DEGAP** | 68.87 | 62.17 | **96.42** | **75.60** | **71.76** | **65.68** | **91.15** | **76.35** | **70.98** | 64.72 | **92.25** | **76.07** |

*Table 4.* **Latency(s) Comparison.** Measured on 500 random MSCOCO samples (Mean$(\pm$Std)).

| Method | LLaVA-1.5 | InstructBLIP | mPLUG-Owl2 |
|---|---|---|---|
| Regular | 3.09 $(\pm 0.18)$ | 3.21 $(\pm 0.31)$ | 5.48 $(\pm 0.24)$ |
| VCD | 6.41 $(\pm 0.25)$ | 6.41 $(\pm 0.34)$ | 11.06 $(\pm 0.50)$ |
| M3ID | 5.97 $(\pm 0.32)$ | 6.23 $(\pm 0.32)$ | 11.33 $(\pm 0.73)$ |
| RITUAL | 6.33 $(\pm 0.44)$ | 6.67 $(\pm 0.30)$ | 11.21 $(\pm 0.55)$ |
| ICD | 5.97 $(\pm 0.31)$ | 6.11 $(\pm 0.35)$ | 11.18 $(\pm 0.35)$ |
| DeGF | 11.86 $(\pm 0.68)$ | 16.00 $(\pm 1.66)$ | 20.07 $(\pm 1.31)$ |
| **DEGAP** | 5.09 $(\pm 0.32)$ | 5.19 $(\pm 0.38)$ | 8.25 $(\pm 0.40)$ |

for DEGAP against each decoding baseline on two backbones. On LLaVA-1.5, DEGAP improves both Acc. and Det. over standard decoding, and continues to outperform other decoding-based mitigation methods under the same evaluation protocol, suggesting that DEGAP enhances overall response quality while remaining practically applicable.

## 4.3. Inference Latency Comparison

We compare the end-to-end inference latency of contrastive decoding methods, measured from input preprocessing to the generation of the final output token including image preprocessing, tokenization, decoding, and post-processing. All method-specific overheads (e.g., input transformation, extra branch construction, and contrastive image generation) are included in this end-to-end measurement. Latency is measured on 500 randomly sampled MSCOCO examples under the same evaluation setting as CHAIR, after three warm-up runs. The results are reported in Table 4.

Contrastive decoding typically increases latency because it requires additional inference to obtain a contrastive distribution, often via modified inputs (e.g., adding noise to the image, generating a synthesized image). Some methods further amplify the overhead when the contrastive branch depends on an explicit input generation step before inference. In contrast, our method constructs contrastive logits via partial inference inside the decoder. We reuse the hidden state from layer $(l_t^* - 1)$ computed in the original branch, and apply visual attention masking only from $l_t^*$ onward. This eliminates a full extra forward pass and recomputes only the later layers, confining the overhead to the post-masking portion of the decoder. As a result, our method yields lower latency overhead than prior contrastive decoding methods while keeping the deployment cost practical for real systems. Additional efficiency results and a detailed analysis of the computational overhead are provided in Appendix B.6.

## 4.4. Ablation Study

**Ablation on Layer Selection Strategy**  In this section, we conduct an ablation study to examine the benefit of our dynamic entropy-guided layer selection over static layer choices. We compare (i) a static strategy that fixes the

*Table 5.* **Results on the LLaVA-Bench.** Higher Accuracy and Detailedness ($\uparrow$) indicate better alignment with image content and richer descriptive quality.

| Method | LLaVA-1.5 | | InstructBLIP | |
|---|---|---|---|---|
| | Acc. $\uparrow$ | Det. $\uparrow$ | Acc. $\uparrow$ | Det. $\uparrow$ |
| Regular | 5.68 | 5.62 | 5.95 | 5.49 |
| **DEGAP** | **6.77** | **6.14** | **7.21** | **5.65** |
| RITUAL | 6.97 | 6.38 | 6.23 | 5.39 |
| **DEGAP** | **7.02** | **6.43** | **7.27** | **6.06** |
| DeGF | 6.75 | 6.23 | 6.30 | 5.32 |
| **DEGAP** | **6.93** | **6.31** | **6.85** | **5.73** |

*Table 6.* **Layer-wise CHAIR results** with $C_S$ (CHAIRs) and $C_I$ (CHAIRi).

| Layer | LLaVA-1.5 | | InstructBLIP | | mPLUG-Owl2 | |
|---|---|---|---|---|---|---|
| | $C_S \downarrow$ | $C_I \downarrow$ | $C_S \downarrow$ | $C_I \downarrow$ | $C_S \downarrow$ | $C_I \downarrow$ |
| 0 | 25.6 | 8.2 | 31.4 | 9.8 | 25.6 | 8.4 |
| 5 | 28.6 | 9.1 | 40.6 | 12.7 | 24.0 | 8.2 |
| 10 | 25.2 | 8.4 | 35.2 | 10.6 | 22.2 | 7.9 |
| 15 | 22.6 | 7.6 | 25.2 | 8.8 | 21.6 | 7.6 |
| 20 | 21.0 | 6.6 | 24.2 | 8.5 | 22.2 | 7.9 |
| 25 | 22.8 | 7.9 | 26.4 | 9.7 | 22.6 | 8.1 |
| 30 | 23.6 | 7.6 | 28.4 | 10.1 | 26.2 | 8.4 |
| **DEGAP** | **19.1** | **6.3** | **22.1** | **7.7** | **20.2** | **7.2** |

starting layer for visual attention masking when constructing contrastive logits, and (ii) our proposed strategy that automatically detects the first descending knee point $l^*$ from the token-wise layer-wise entropy curve. For the static layer selection, we sweep the masking start layer over $l \in \{0, 5, 10, 15, 20, 25, 30\}$ and evaluate CHAIR under the same decoding configuration and evaluation protocol.

Table 6 shows that static strategy is highly sensitive to the chosen layer. When masking starts too early, visual information is blocked before multimodal alignment is sufficiently formed, causing the contrastive logits to drift toward generic noise rather than exhibiting hallucination tendencies. Conversely, when masking starts too late, visual evidence has already been largely integrated into the hidden states, making masking ineffective and the contrastive logits overly similar to the original ones. As a result, static interventions yield improvements only in a narrow range of layers. Additional sensitivity analyses on the contrastive strength and entropy-based layer selection strategy are provided in Appendix B.8.

In contrast, our dynamic layer selection mitigates this sensitivity. Specifically, it identifies, at each decoding step, the layer boundary where the model starts forming strong next-token confidence. By applying perturbation starting from this boundary, the resulting contrastive logits more consistently reflect hallucination patterns. This leads to more stable and improved CHAIR performance. Empirically, the

proposed layer selection achieves the best CHAIR score among all tested static layer settings and substantially reduces the performance variability observed with fixed-layer choices.

*Table 7.* Effect of visual token masking ratio on LLaVA-1.5-7B.

| Masking Ratio | $C_S \downarrow$ | $C_I \downarrow$ | Rec. $\uparrow$ | Len. $\uparrow$ |
|:---:|:---:|:---:|:---:|:---:|
| 20% | 26.0 | 8.2 | 59.1 | 54.1 |
| 40% | 24.4 | 8.2 | 60.9 | 54.4 |
| 60% | 23.2 | 7.9 | 62.0 | 54.4 |
| 80% | 22.4 | 8.4 | 60.8 | 54.3 |
| 100% (DEGAP) | **19.1** | **6.3** | **65.2** | **54.5** |

**Effect of Visual Token Masking Ratio** To further examine the design of the contrast branch, we also conduct an ablation study on the ratio of visual tokens masked after $l^*$. As shown in Table 7, performance improves as the masking becomes stronger, and full masking achieves the best overall result. We attribute this to the fact that partial masking leaves residual visual signals in the contrast branch, weakening the separation between the original and contrastive logits. In contrast, full masking more clearly suppresses visual grounding in the contrast branch, making hallucination-prone language priors more explicitly exposed.

## 5. Conclusion

In this work, we propose DEGAP, a contrastive decoding method that enhances visual grounding without additional training or external input perturbations, to address hallucinations in LVLMs. We analyze the effect of attention perturbations on VLM outputs across layers and observe that their impact becomes more pronounced in the later layers. We further find that this pattern aligns closely with the model's confidence as reflected by the layer-wise logit entropy. Based on this finding, we introduce dynamic entropy-guided layer selection, which determines the attention perturbation point at each decoding step. DEGAP efficiently computes contrastive logits by performing partial inference only after the selected layer, enabling faster contrastive decoding than existing methods. Experimental results show that DEGAP consistently outperforms existing approaches across hallucination benchmarks and general tasks.

## Acknowledgements

This work was supported by the Institute of Information and Communications Technology Planning and Evaluation (IITP) grants (No. RS-2025-25422680 and No. RS-2020-II201373), and by the National Research Foundation of Korea (NRF) grant (No. RS-2025-00520618), funded by the Korean Government (MSIT).

## Impact Statement

DEGAP provides a novel, training-free approach to mitigate hallucinations in Large Vision–Language Models (LVLMs) by dynamically perturbing visual attention during decoding. Unlike prior methods that rely on input-level perturbations or extra inference, DEGAP directly reflects the model's internal reliance on visual signals, ensuring more accurate and visually grounded outputs. Its entropy-guided dynamic layer selection enables efficient and targeted interventions, reducing computational overhead while improving robustness across diverse vision-language tasks. DEGAP's lightweight and broadly applicable design makes it a practical solution for enhancing the reliability of LVLMs in applications such as visual question answering, image captioning, and multimodal reasoning, where hallucination-free outputs are critical.

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

# A. Experimental Details

## A.1. Benchmarks and Metrics

**CHAIR** (Rohrbach et al., 2018) evaluates object hallucinations in open-ended captioning tasks. It prompts the LVLMs to describe specific images selected from a random sample of 500 images from the MSCOCO (Lin et al., 2014) validation set and assesses performance based on two metrics from (Rohrbach et al., 2018)

$$\text{CHAIR}_I = \frac{\text{\# hallucinated objects}}{\text{\# all objects mentioned}} \tag{10}$$

$$\text{CHAIR}_S = \frac{\text{\# sentences with hallucinated object}}{\text{\# all sentences}} \tag{11}$$

**POPE** (Li et al., 2023b) presents models with binary (yes/no) questions that ask about the presence of specific objects. The benchmark utilizes data from three well-known datasets: MSCOCO, A-OKVQA (Schwenk et al., 2022), and GQA (Hudson & Manning, 2019). Based on the strategy for generating negative samples, the benchmark is divided into three subsets: random, popular, and adversarial. For each dataset configuration, six questions are asked per image (total of 3,000 test instances). To assess model performance, we employ four evaluation metrics: accuracy, precision, recall, and F1 score.

**MME benchmark** (Fu et al., 2023) provides a comprehensive evaluation of LVLMs by dividing hallucination assessment into four subsets: object-level (existence and count) and attribute-level (position and color). Each subset contains 30 images, with two yes/no questions per image—totaling 60 questions. Like POPE, all questions are binary in nature, and model performance is measured using binary accuracy.

**MMBench** (Liu et al., 2024b) evaluates the multimodal understanding and reasoning skills of AI models. It emphasizes scenarios that demand the fusion of visual and textual inputs. The benchmark adopts a hierarchical taxonomy of abilities, with Perception and Reasoning defined as Level-1 (L-1) categories. This structure is further expanded by breaking down these categories into six Level-2 (L-2) and twenty Level-3 (L-3) dimensions.

**LLaVA-Bench** (Liu et al., 2023b) includes 24 images that span a variety of complex visual types, including scenes, memes, paintings, and sketches, accompanied by 60 difficult questions. From this dataset, we sample representative examples to qualitatively compare the outputs produced by different decoding strategies. We assess the accuracy and richness of the generated responses using the advanced GPT-4o.

**MM-Vet** (Yu et al., 2023) evaluates the overall performance of multimodal models across a wide range of vision-and-language tasks. It includes a diverse set of questions consisting of 16 tasks and 218 instances such as object recognition, counting, commonsense reasoning, and spatial understanding. The score is measured by `https://huggingface.co/spaces/whyu/MM-Vet_Evaluator`

**HallusionBench** (Guan et al., 2024) is designed to evaluate hallucination in LVLMs. It contains diverse image–question pairs spanning real-world objects, attributes, and commonsense scenarios. Each instance is evaluated through binary (yes/no) and multiple-choice questions, enabling quantitative assessment of hallucination frequency and severity. Performance is typically reported using accuracy and hallucination rate.

## A.2. Query Prompt

We provide the prompt templates used in each benchmark, including the generation prompt used to elicit the model's response, as well as the evaluation prompt used during assessment.

For the POPE benchmark, the generation prompt is *"Is there a {object} in the image?"*. In the case of MME, the generation prompt is *"Is there a {object} in the image? Answer the question using a single word or phrase."* For CHAIR, the generation prompt is *"Please describe this image in detail."* Finally, for the LLaVA-Bench, we present the prompt provided to GPT-4o in Figure 7, which is used to evaluate the responses of large vision-language models (LVLMs) based on both accuracy and detail.

## A.3. Details of Other Baselines

In this study, we compare the performance of DEGAP with seven recent state-of-the-art approaches: VCD (Leng et al., 2024), M3ID (Favero et al., 2024), RITUAL (Woo et al., 2024), ICD (Wang et al., 2024), DeGF (Zhang et al., 2025), OPERA

(Huang et al., 2024), and HALC (Chen et al., 2024). The details and implementation of these methods are as follows.

**VCD** (Leng et al., 2024) takes as input a text query $x$ and a visual input $v$, and generates two conditional output distributions: one conditioned on the original image $v$, and the other on a distorted version $v'$, created using a predefined transformation (e.g., Gaussian noise masking). By contrasting these two distributions, VCD computes a contrastive probability as follows:

$$p_\theta(y_t) = \mathrm{Softmax}\Big[(1 + \alpha)f_\theta(y \mid v, \mathbf{x}, \mathbf{y}_{<t}) - \alpha f_\theta(y \mid v', \mathbf{x}, \mathbf{y}_{<t})\Big] \tag{12}$$

In our implementation, we follow the default settings of VCD by setting $\alpha = 1$, and obtain $v'$ through a 500-step noise injection process.

**M3ID** (Favero et al., 2024) compares the distribution conditioned on the image with a text-only conditional distribution that excludes visual input. The final output probability $p_{m3id}(y_t)$ is defined as follows:

$$p_\theta(y_t) = \mathrm{Softmax}\Big[ f_\theta\big(y \mid v, \mathbf{x}, \mathbf{y}_{<t}\big) + \frac{1 - e^{-\lambda t}}{e^{-\lambda t}}\Big(f_\theta\big(y \mid v, \mathbf{x}, \mathbf{y}_{<t}\big) - f_\theta\big(y \mid \mathbf{x}, \mathbf{y}_{<t}\big)\Big)\Big] \tag{13}$$

Following the recommended configuration, we fix $\lambda = 0.02$, which serves as a hyperparameter to balance the influence between the vision-conditioned model and the text-only model.

**RITUAL** (Woo et al., 2024) applies common transformations such as cropping, flipping, and color jitter to the original image $v$ to generate a transformed image $v^{(T)}$, and utilizes both images simultaneously to generate responses. The resulting probability distribution is defined as follows:

$$p_\theta(y_t) = \mathrm{Softmax}\Big[f_\theta\big(y \mid v, \mathbf{x}, \mathbf{y}_{<t}\big) + \kappa\, f_\theta\big(y \mid v^{(T)}, \mathbf{x}, \mathbf{y}_{<t}\big)\Big] \tag{14}$$

The contribution of the transformed image is controlled by a hyperparameter $\kappa = 3$, which follows the value used in the official implementation.

**ICD** (Wang et al., 2024) is a contrastive approach that compares the distribution derived from the original instruction $X_{ins}$ with that of a disturbance instruction $X'_{ins}$. The disturbance instructions are generated by prepending role prefixes to the original instruction, which increases multimodal alignment uncertainty and exacerbates hallucinations. ICD employs a "highlight-then-separate" strategy to identify and remove hallucination-prone components from the original distribution. The objective function of ICD is defined as follows:

$$p_\theta(y_t) = \mathrm{softmax}\Big[f_\theta\big(y_t \mid X_V, X_{\mathrm{ins}}, y_{<t}\big) - \lambda f_\theta\big(y_t \mid X_V, X'_{\mathrm{ins}}, y_{<t}\big)\Big] \tag{15}$$

We set $\lambda = 1$ according to the official implementation to control the contribution of $X'_{ins}$.

**DeGF** (Zhang et al., 2025) proposes a self-correcting decoding method using generative feedback. At each timestep, the Jensen-Shannon divergence $d_t(v, v')$ between token distributions conditioned on the original image $v$ and a generated reference $v'$ determines whether to confirm or revise the prediction:

$$p_\theta(y_t) = \begin{cases} \mathrm{Softmax}[\, f_\theta(y_t \mid v, \mathbf{x}, \mathbf{y}_{<t}) + \alpha_1 f_\theta(y_t \mid v', \mathbf{x}, \mathbf{y}_{<t})], & \text{if } d_t(v, v') < \gamma; \\ \mathrm{Softmax}[\, (1 + \alpha_2)f_\theta(y_t \mid v, \mathbf{x}, \mathbf{y}_{<t}) - \alpha_2 f_\theta(y_t \mid v', \mathbf{x}, \mathbf{y}_{<t})], & \text{if } d_t(v, v') \geq \gamma \end{cases} \tag{16}$$

Following the official implementation, we set $\alpha_1 = 3$ and $\alpha_2 = 1$.

**OPERA** (Huang et al., 2024) introduces a decoding method to reduce hallucinations in multimodal large language models by combining Over-Trust Logit Penalty and Retrospection-Allocation. It first penalizes tokens with overly concentrated attention by adjusting their selection probabilities as follows:

$$p_\theta(x_t \mid x_{<t}) = \mathrm{Softmax}\left(H(h_t) - \alpha\,\phi(w_{<t})\right) \tag{17}$$

Here, $\phi(w_{<t})$ is a penalty term derived from attention concentration, and $\alpha$ controls the strength of this penalty; in experiments, $\alpha$ is set to 1. In addition, when recent tokens repeatedly attend to the same location, indicating a potential hallucination, OPERA rolls back the decoding to that point and regenerates subsequent tokens to reduce the likelihood of hallucinated content.

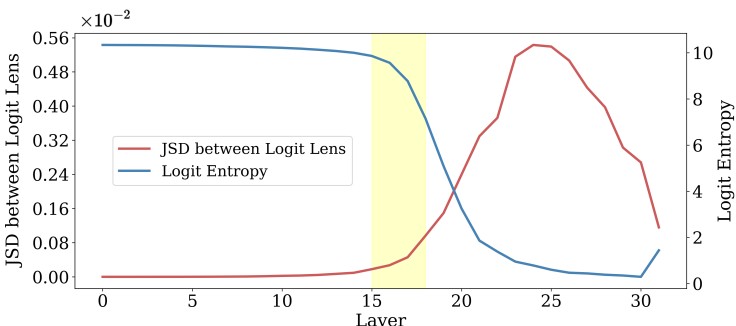

*Figure 5.* Layer-wise perturbation effects and logit entropy in LVLM outputs. Jensen–Shannon Divergence (JSD) between token distributions from the original and perturbed images measured by a logit lens at each layer, and the original-branch logit-lens entropy.

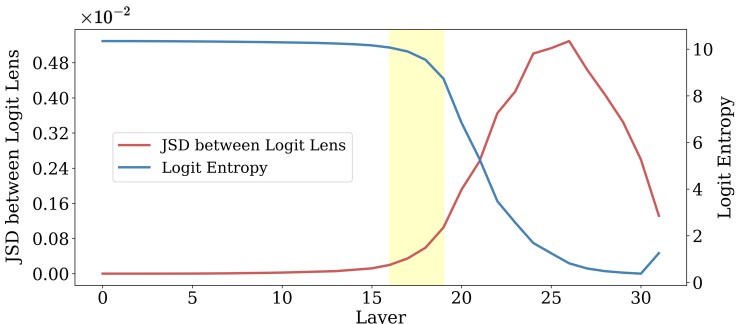

*Figure 6.* Layer-wise perturbation effects and logit entropy in LVLM outputs. Jensen–Shannon Divergence (JSD) between token distributions from the original and perturbed images measured by a logit lens at each layer, and the original-branch logit-lens entropy.

**HALC** (Chen et al., 2024) dynamically adjusts the probability distribution of each generated token based on fine-grained visual contexts. Specifically, given a token $y_t$, it samples a set of $n$ extended visual contexts $\{v_1, ..., v_n\}$ centered around a primary visual context $v_d$. It then measures the divergence between decoding distributions for each pair of visual contexts $(v_i, v_j)$ using the Jensen-Shannon Divergence (JSD). The top $m$ field-of-view (FOV) pairs are selected, and a contrastive probability distribution is computed as follows:

$$p_\theta(y_t) = \text{Softmax}\Big[(1 + \alpha)\, f_\theta\left(y \mid v_i, \mathbf{x}, \mathbf{y}_{<t}\right) - \alpha\, f_\theta\left(y \mid v_j, \mathbf{x}, \mathbf{y}_{<t}\right)\Big] \tag{18}$$

We set the number of visual context samples to $n = 4$, the number of contrastive candidate pairs to $m = 6$, and the emphasis coefficient to $\alpha = 0.05$ according to the official implementation.

**DROPOUT** (Fang et al., 2024) is an inference-time decoding method that projects visual tokens into the text vocabulary space to quantify per-token uncertainty, then decodes in parallel under multiple masked contexts where tokens with high epistemic uncertainty are probabilistically dropped, and finally selects the next token by majority vote. Concretely, for each visual token $x_i^v$, the top-layer decoder hidden representation is projected into the text space to obtain a distribution $q_i^{proj}$, and the epistemic uncertainty is measured as the KL divergence to the mean projection $q^{proj}$, i.e., $U_{\text{epi}}(i) = \text{D}_{\text{KL}}\Big(q_i^{\text{proj}} \parallel q^{\text{proj}}\Big)$.

Dropout probabilities are then determined from this value (after linear normalization and strength control with $\gamma^{(k)}$ and $\delta^{(k)}$), and $K$ visual-token masks $M^{(k)}$ are sampled. Candidate predictions obtained under each mask are aggregated by majority vote. In our experimental setup, we use $K = 3$, which follows the value used in the official implementation.

## B. Additional Results

### B.1. Additional Analysis

We extend the layer-wise attention-perturbation analysis in Section 3.2 and Section 3.3 from LLaVA to InstructBLIP and mPLUG-Owl2. We use the same experimental setup as Section 3.2 and follow the same analysis protocol as Section 3.3, computing the layer-wise Jensen–Shannon divergence (JSD) between the logit-lens distributions from the original branch and the perturbed branch with visual attention masked, together with the output entropy computed from the original branch's

*Table 8.* Results on **MMBench** with six Level-2 (L-2) categories: Logical Reasoning (LR), Attribute Reasoning (AR), Relation Reasoning (RR), Fine Perception-Single (FP-S), Fine Perception-Cross (FP-C), and Coarse Perception (CP).

| Model | LR ↑ | AR ↑ | RR ↑ | FP-S ↑ | FP-C ↑ | CP ↑ | Overall ↑ |
|---|---|---|---|---|---|---|---|
| Regular | 0.1525 | 0.4271 | 0.3565 | 0.3890 | 0.3216 | 0.5472 | 0.4003 |
| ICD | 0.1525 | 0.4924 | 0.4869 | 0.4266 | 0.4335 | 0.5641 | 0.4518 |
| VCD | 0.2457 | 0.6532 | 0.5043 | 0.5426 | 0.4895 | 0.7364 | 0.5704 |
| M3ID | 0.2203 | 0.6030 | 0.5043 | 0.5290 | 0.4755 | 0.7094 | 0.5472 |
| RITUAL | 0.2457 | 0.6631 | 0.5043 | 0.5460 | 0.4965 | 0.7364 | 0.5738 |
| DeGF | **0.2796** | **0.7331** | **0.5304** | 0.5972 | 0.5524 | 0.7331 | 0.6048 |
| **DEGAP** | 0.2711 | 0.6936 | **0.5304** | **0.6388** | **0.5805** | **0.7570** | **0.6243** |

*Table 9.* Results on **MM-Vet** categories: Recognition (Rec), Optical Character Recognition (OCR), Knowledge (Know), Language Generation (Gen), Spatial Awareness (Spat), Mathematics (Math).

| Method | Rec ↑ | OCR ↑ | Know ↑ | Gen ↑ | Spat ↑ | Math ↑ | Total ↑ |
|---|---|---|---|---|---|---|---|
| Regular | 28.7 | 14.4 | 13.5 | 14.5 | 18.0 | 5.4 | 22.9 |
| VCD | 27.5 | 20.3 | 11.4 | 12.4 | 20.5 | 11.5 | 25.5 |
| ICD | 30.1 | 14.2 | 15.5 | 15.1 | 24.3 | 7.7 | 24.7 |
| M3ID | 29.6 | 20.4 | 15.1 | 13.6 | **26.1** | 11.5 | 26.7 |
| RITUAL | 32.1 | 17.5 | 16.2 | 15.1 | 23.5 | 7.7 | 27.4 |
| DeGF | 31.5 | 17.9 | 13.6 | **16.0** | 20.9 | 7.7 | 27.1 |
| **DEGAP** | **32.9** | **21.4** | **17.0** | 15.0 | **26.1** | **15.4** | **29.4** |

logit-lens distributions.

Figure 5 presents the results for InstructBLIP. JSD stays near zero in early layers and becomes substantially larger only in later layers. The rise in JSD aligns with the stage where the layer-wise output entropy starts to decrease, indicating that the attention-masking perturbation most strongly affects the output distribution when predictions begin to sharpen.

Figure 6 shows analogous trends for mPLUG-Owl2. Perturbation sensitivity is small in early layers but grows in the mid-to-late layers, and this growth again coincides with the entropy drop rather than appearing uniformly across layers.

Overall, we observe the same phenomenon across multiple LVLMs: perturbation sensitivity (JSD) rises as predictions sharpen (entropy decreases) across layers. The consistent emergence of this trend in diverse LVLMs indicates that the entropy-based layer-selection signal generalizes well.

### B.2. Full Results on MMBench

As shown in Table 8, DEGAP achieves the highest Overall score, surpassing the previous best method. DeGF remains competitive in LR and AR, but lags in fine-perception and coarse-perception categories. Overall, DEGAP delivers the most balanced gains across reasoning and perception tasks on MMBench. The "Regular" setting corresponds to the original LLaVA-1.5 7B.

### B.3. Full Results on MM-Vet

Table 9 shows MM-Vet scores with Regular LLaVA-1.5 as baseline. DEGAP tops all methods. Improvements are most striking in Math (surpassed the regular model by 185%) and OCR (4.9% over M3ID), proving robustness on equations and text recognition. Knowledge also rises, showing hallucination control does not impair factual reasoning. These uniform gains across every MM-Vet category demonstrate that our approach curbs hallucinations while boosting recall in varied reasoning tasks, offering high reliability for real-world applications such as document understanding and data-visualization queries.

### B.4. Scaling Up LVLMs

To evaluate the generalization capability of DEGAP, we further report additional POPE results on larger-scale models and different VLM architectures. These experiments were conducted on the MSCOCO dataset. As shown in Table 10, on LLaVA-1.5 13B, DEGAP achieved 88.24% Accuracy and 89.08% F1, attaining the best performance. On another VLM

*Table 10.* Results on POPE for **LLaVA-1.5 13B**, **Qwen2.5-VL 7B**, and **Qwen2.5-VL 32B**.

| Method | LLaVA-1.5 13B | | Qwen2.5-VL 7B | | Qwen2.5-VL 32B | |
|---|---|---|---|---|---|---|
| | Acc. ↑ | F1 ↑ | Acc. ↑ | F1 ↑ | Acc. ↑ | F1 ↑ |
| Regular | 83.64 | 84.95 | 83.13 | 80.63 | 90.58 | 90.24 |
| VCD | 83.75 | 85.22 | 82.13 | 80.74 | 90.62 | 90.30 |
| M3ID | 84.71 | 85.98 | 81.99 | 80.64 | 90.62 | 90.30 |
| ICD | 84.78 | 86.02 | 82.16 | 80.70 | 89.24 | 88.58 |
| RITUAL | 87.53 | 88.42 | 81.79 | 80.28 | 89.73 | 89.58 |
| DeGF | 86.80 | 86.87 | 82.82 | 80.89 | 91.41 | 91.10 |
| **DEGAP** | **88.24** | **89.08** | **85.13** | **81.20** | **91.60** | **91.28** |

architecture, Qwen2.5-VL 7B, DEGAP also outperformed all baselines, and when scaling up to Qwen2.5-VL 32B, it reached 91.60% Accuracy and 91.28% F1, again yielding the highest results. These findings demonstrate that DEGAP generalizes well across both architectures and model scales.

### B.5. Results on Recent High-Performance LVLMs

To further assess the generalizability of DEGAP, we evaluate our method on two recently released open-source LVLMs: Qwen3-VL-8B and InternVL3.5-8B. As shown in Table 11, DEGAP consistently achieves the lowest hallucination scores on CHAIR across both architectures, demonstrating that the benefits of our method generalize beyond the models used in the main experiments.

*Table 11.* Results on CHAIR for Qwen3-VL-8B and InternVL3.5-8B.

| Method | Qwen3-VL-8B | | InternVL3.5-8B | |
|---|---|---|---|---|
| | $C_S \downarrow$ | $C_I \downarrow$ | $C_S \downarrow$ | $C_I \downarrow$ |
| Regular | 12.0 | 6.9 | 21.2 | 8.6 |
| VCD | 11.6 | 6.3 | 20.8 | 8.4 |
| M3ID | 11.6 | 6.4 | 20.4 | 8.7 |
| DeGF | 10.8 | 6.0 | 20.1 | 8.2 |
| DEGAP | **9.2** | **5.2** | **18.6** | **7.1** |

### B.6. Efficiency Comparison

Table 12 summarizes the inference efficiency of DEGAP on CHAIR with LLaVA-1.5-7B. We report wall-clock latency, end-to-end TFLOPs per sample, and peak GPU memory usage under the same experimental setting. Although DEGAP, like other contrastive decoding methods, introduces additional inference cost relative to regular decoding, it avoids a full second forward pass by reusing hidden states before the masking start layer and performing partial inference only on the remaining layers. This design yields a more favorable efficiency–performance trade-off than existing contrastive decoding baselines. As shown in Table 12, DEGAP achieves the lowest latency and the lowest computational cost among the compared contrastive methods, while its peak GPU memory usage remains close to that of regular decoding.

*Table 12.* Efficiency comparison on CHAIR with LLaVA-1.5-7B.

| Method | ms/token ↓ | TFLOPs/sample ↓ | GPU peak memory (GB) ↓ |
|---|---|---|---|
| Regular | 53.92 | 9.72 | 14.5 |
| VCD | 104.41 | 19.28 | 15.1 |
| M3ID | 104.02 | 11.20 | 14.9 |
| DeGF | 210.28 | 29.17 | 17.7 |
| DEGAP | 88.01 | 12.96 | 14.6 |

## B.7. Additional Comparison with DeCo

We further compare DEGAP with DeCo (Wang et al., 2025), a closely related decoding-time method that also performs layer-wise analysis during inference. Although both methods intervene during decoding and exploit internal model signals, their objectives and mechanisms are fundamentally different. DeCo seeks a reliable intermediate layer whose representations preserve visual evidence and then corrects the final prediction by reinforcing logits from that layer. In contrast, DEGAP identifies the layer from which visual information should be perturbed so that the contrast branch can better expose hallucination tendencies, and then suppresses hallucination-prone tokens by contrasting the perturbed branch with the original branch. To clarify this distinction, we conduct comparisons with DeCo on both LLaVA-1.5 7B and Qwen3-VL 8B. Table 13 reports the results on CHAIR and POPE. DEGAP consistently outperforms DeCo across both model architectures, demonstrating that our method is more effective for hallucination mitigation.

*Table 13.* Comparison with DeCo on **LLaVA-1.5 7B** and **Qwen3-VL 8B**.

| Method | LLaVA-1.5 7B | | | | Qwen3-VL 8B | | | |
|---|---|---|---|---|---|---|---|---|
| | $C_S \downarrow$ | $C_I \downarrow$ | Acc. $\uparrow$ | F1 $\uparrow$ | $C_S \downarrow$ | $C_I \downarrow$ | Acc. $\uparrow$ | F1 $\uparrow$ |
| DeCo | 21.4 | 7.6 | 87.83 | 88.19 | 9.8 | 5.5 | 88.88 | 89.21 |
| **DEGAP** | **19.2** | **6.3** | **88.93** | **89.22** | **9.2** | **5.2** | **89.42** | **89.29** |

In addition, we analyze the distribution of selected layers for both methods on CHAIR. As shown in Table 14, DeCo mainly selects deeper layers, whereas DEGAP more frequently selects earlier-to-middle layers. This observation further supports that the two methods perform layer selection for different purposes.

*Table 14.* Distribution of selected layers on CHAIR with Qwen3-VL-8B.

| layer | $\leq 18$ | 19 | 20 | 21 | 22 | 23 | 24 | 25 | 26 | 27 | 28 | $\geq 29$ |
|---|---|---|---|---|---|---|---|---|---|---|---|---|
| DeCo | 0.0 | 0.0 | 1.8 | 2.6 | 3.8 | 6.1 | 9.6 | 7.8 | 14.7 | 20.7 | 33.0 | 0.0 |
| DEGAP | 34.7 | 20.8 | 24.8 | 14.3 | 3.2 | 0.3 | 0.1 | 0.2 | 0.2 | 0.3 | 0.1 | 1.1 |

## B.8. Sensitivity Analysis

We conduct two additional sensitivity analyses to examine the robustness of DEGAP with respect to its design choices.
**Effect of Contrastive Strength $\alpha$.** In DEGAP, $\alpha$ determines the contribution of the contrastive logits to the final prediction, and we use $\alpha = 1$ for all tasks following prior contrastive decoding methods for fair comparison. To verify that DEGAP is not sensitive to this choice, we evaluate CHAIR performance across multiple $\alpha$ values on LLaVA-1.5-7B. As shown in Table 15, DEGAP outperforms regular decoding across all values, indicating that our method remains effective over a reasonably broad range of $\alpha$ rather than depending on a narrowly tuned setting.

**Effect of Layer Selection Strategy.** Beyond fixed-layer settings, we also evaluate a simple entropy-threshold heuristic that selects the first layer below a preset threshold. As shown in Table 15, its performance varies with the threshold and remains below DEGAP across all tested values. Unlike this approach, DEGAP maintains strong robustness across diverse settings without additional tuning.

*Table 15.* Sensitivity analysis of $\alpha$ and effect of entropy threshold on CHAIR with LLaVA-1.5-7B.

| Varying $\alpha$ | | | Varying entropy threshold | | |
|---|---|---|---|---|---|
| $\alpha$ | $C_S \downarrow$ | $C_I \downarrow$ | Threshold | $C_S \downarrow$ | $C_I \downarrow$ |
| 0 (Regular) | 27.2 | 9.8 | 8.4 | 23.0 | 7.6 |
| 0.25 | 21.0 | 7.4 | 8.8 | 23.4 | 7.7 |
| 0.50 | 21.2 | 7.5 | 9.2 | 23.4 | 8.0 |
| 0.75 | 20.8 | 7.5 | 9.6 | 23.0 | 8.1 |
| 1.0 (Ours) | **19.1** | **6.3** | DEGAP | **19.1** | **6.3** |

## B.9. Qualitative Results

Figure 8 and Figure 9 qualitatively demonstrate the hallucination mitigation effect of our method on the CHAIR and LLaVA-Bench benchmarks, respectively. All results were generated using the LLaVA-1.5 model. In Figure 8, the regular model produces hallucinated objects such as wine glass and fork, while our method generates responses grounded in actual visual content, significantly reducing hallucinations. In Figure 9, our method provides more accurate and detailed answers to image-based queries, achieving higher Accuracy and Detailedness scores in GPT-4o-based evaluation compared to the regular model. The prompts used for GPT-4o-based evaluation are provided in Figure 7.

---

**Description:**
AI that scores image description accuracy and detailedness.

---

**Instructions:**
You are an AI designed to evaluate and score the performance of two AI assistants in describing a given image. Your primary focus is on the accuracy and detailedness of their descriptions. You will assess the accuracy by checking for hallucinations - any part of the description that is inconsistent with the image content. For detailedness, you will consider how rich the response is in necessary details, excluding any hallucinated parts. You will provide scores on a scale from 1 to 10 for each assistant separately, based on these criteria. After scoring, you will offer an explanation for your evaluation, ensuring it is free from bias and not influenced by the order of presentation of the responses.

Input format:

[Assistant 1]
{Response 1}
[End of Assistant 1]

[Assistant 2]
{Response 2}
[End of Assistant 2]

Output format:

Accuracy:
Scores of the two answers:
Reason:

Detailedness:
Scores of the two answers:
Reason:

---

*Figure 7.* The prompt given to GPT-4o for scoring LVLM outputs on accuracy and detailedness.

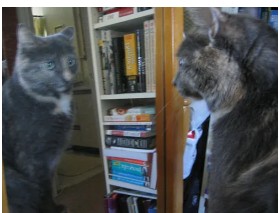 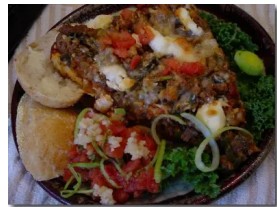 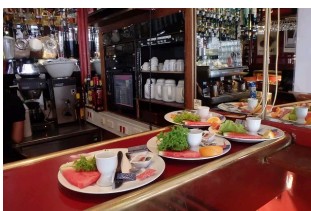

**Regular:** The image features two cats staring into each other's reflection in a mirror. Both cats appear to be curious, as they gaze at each other through the mirror's surface. The scene takes place in a room with various objects and furniture around. On a shelf, there is a collection of books, and a cup can be seen as well. A remote control and a tie are placed close to it, and scissors are also visible near the cup. There is a chair positioned in the room, near the right side of the scene. Additionally, a pair of scissors is placed on the floor. In the foreground, the cats are the main focus of the scene, capturing attentive viewers' attention.

**Ours:** The image features two cats sitting in a room, facing each other and looking at each other through a mirror. The room is filled with numerous books scattered around, both on the floor and on shelves. The cats are positioned near the center of the room, with one cat on the left side and the other on the right side. The mirror reflects the cats' images, creating a unique perspective of the scene.

**Regular:** The image displays a delicious and colorful meal served in a bowl. The main dish features a mix of cheesy meat, accompanied by chunks of broccoli and tomatoes arranged around various parts of the bowl. There are multiple slices of broccoli in the dish, both on the top and below the main dish. A couple of strips of red onion can be seen as well. To enhance the presentation, there are a few carrots placed both around and maybe even on the side of the dish. Two crusty biscuits serve as a side to the main dish, further complementing the meal with crunchy texture. Two cups are visible in the background, one on the left and another at the top, implying there might be drinks individually served for each person as they enjoy the meal.

**Ours:** The image features a delicious meal served on a plate, consisting of a variety of food items. There is a pizza with cheese and tomatoes, accompanied by a side of vegetables, including broccoli and a slice of cucumber. The plate also includes a portion of meat, likely beef, and a roll. The dish appears to be a hearty and flavorful combination of ingredients, making it an appetizing meal.

**Regular:** The scene features a long, red-covered counter with a salad bar filled with a variety of plates and bowls containing a combination of berries, cherry tomatoes, salad greens, and oranges. Four plates are placed atop different parts of the counter and one large bowl on the far left. There is an impressive array of kitchen appliances throughout the area, including three microwaves, a refrigerator located on the right, a bowl, a number of cups, and a bottle placed near the center of the scene. Additionally, a dining table is present in the background to the right of the bar. The arrangement of the appliances, fruits, and cups creates a fresh, inviting atmosphere for customers to enjoy their salads and other offerings.

**Ours:** The image features a counter with four plates of food, each containing a variety of items. The plates are arranged in a row, with the first plate on the left, the second plate in the middle, the third plate on the right, and the fourth plate at the far right. There are also several cups placed around the counter, with one cup near the leftmost plate, another cup in the middle, and the third cup on the right side. A knife is visible on the counter, likely used for cutting and serving the food. In the background, there are several bottles and a bowl, indicating that the counter is part of a restaurant or dining area.

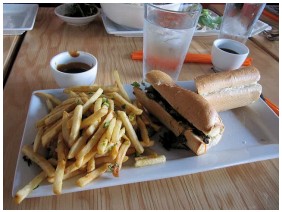 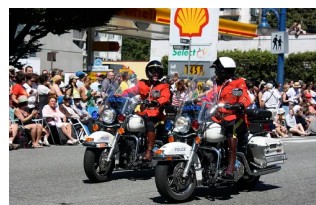 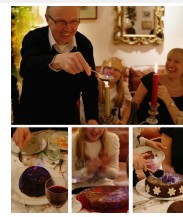

**Regular:** The image features a white plate containing a thick meat sandwich, French fries, and some sauce placed on dining table in a restaurant. The sandwich appears to be a submarine, and the plate is filled with a substantial serving of delicious food. There are two bowls present on the table as well; one next to the sandwich and another positioned further away from the plate. A wine glass is sitting adjacent to the bowl near the plate, possibly for a drink to complement the meal. Additionally, there are a couple of cups—one on the left side and one on the right corner—on the table. A fork can be seen on the right side of the plate, ready for use.

**Ours:** The image features a dining table with a plate of food, including a sandwich and a side of french fries. The sandwich is placed in the center of the plate, while the fries are scattered around it. There are also two bowls on the table, one positioned near the top left corner and the other near the top right corner. In addition to the main dish, there are two cups on the table, one located in the top left corner and the other in the top right corner. A spoon can be seen resting on the table, closer to the top right corner. The table appears to be set for a meal, with the sandwich and fries as the main course.

**Regular:** The image features two uniformed men riding motorcycles in a parade formation. They are both wearing red sashes and police hats, giving the scene an air of organization and processional pride. The first motorcycle is positioned in the foreground, while the second one is trailing behind. Additionally, there are numerous onlookers lining the street, showing their enthusiasm for the parade. Some people are standing near the motorcycles and police officers, while others are scattered further back. In the background, numerous persons can be seen seated or standing on various objects, such as chairs or benches. Numerous teddy bears can also be spotted fair along the street adding charm and vibrance to the parade setting.

**Ours:** The image features two police officers riding motorcycles down a street, likely participating in a parade or event. They are wearing red coats and are positioned next to each other, with one officer slightly ahead of the other. There are several other people in the scene, some of whom are standing or walking along the street, while others are sitting on chairs. Some of the people are closer to the motorcycles, while others are further away. The presence of multiple people and the police officers on motorcycles create a lively atmosphere in the scene.

**Regular:** The image features a man and a woman standing next to each other, engaged in action. The woman is pouring before cutting into a cake, while the man hands her a dessert cake knife for her convenience. They seem to be trying to taste or cook the cake. An extended table can be seen in the background, covered with dishes, glasses, and a variety of food. There are several cups, wine glasses, and additional bottles placed across the table. It effectively showcases the festive environment and warm hospitality surrounding the cake cutting moment.

**Ours:** The image is a collage of four pictures featuring a man and a woman enjoying a festive meal together. In the first picture, the man is holding a spoon over a plate of food, possibly a cake, while the woman smiles. In the second picture, the woman is holding a spoon over a plate of food, and the man is also present in the scene. The third picture shows the couple sitting on a couch, with the woman holding a spoon over a plate of food, and the man is also present. In the fourth picture, the woman is holding a spoon over a plate of food, and the man is sitting next to her. There are multiple wine glasses and cups placed on the dining table, suggesting that they are enjoying a meal with drinks. A cake is also visible in the scene, adding to the festive atmosphere. The couple appears to be having a great time together, sharing a delightful dining experience.

*Figure 8.* **Qualitative CHAIR examples.** Comparison of generated answers for selected images using Regular LLaVA-1.5 vs. DEGAP. The hallucinated objects are shown in red, and ground-truth objects are shown in blue.

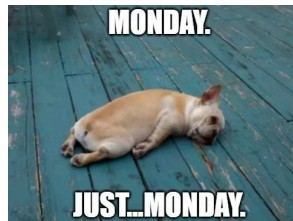

**Query:** Please explain the meme in detail.

**[Regular]**    GPT-4o-Aided Evaluation: Accuracy:4 Detailedness:3

The image shows a puppy dog lying on a blue wooden table on its back, looking up at the camera. The table seems to be a hard surface, like a picnic table, and is where the dog appears to be resting. There is a chair located beside the table, and an empty bottle can be found close to the dog and the chair. The scene is set on what appears to be a wooden deck, possibly on a player's table.

**[Ours]**    GPT-4o-Aided Evaluation: Accuracy:8 Detailedness:6

The image features a small brown dog lying on its back on a wooden floor, appearing to be sleeping or relaxed. The dog is positioned in the middle of the scene, taking up a significant portion of the frame. The image is accompanied by a caption that reads, "Monday. Just… Monday.", which suggests that the dog is enjoying its day off or taking a break from its usual routine. The caption is placed at the bottom of the image, with the dog's position and the wooden floor as the main focus.

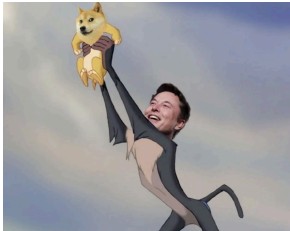

**Query:** Which iconic movie scene is being parodied in the meme?

**[Regular]**    GPT-4o-Aided Evaluation: Accuracy:2 Detailedness:2

The meme is parodying the iconic Iago scene from Walt Disney's Aladdin, where a monkey is holding a baby, making funny facial expressions while raising his child.

**[Ours]**    GPT-4o-Aided Evaluation: Accuracy:9 Detailedness:8

The meme is parodying the iconic movie scene from "The Lion King" where a lion cub is being held up by its father, the Lion King, to show off his newborn cub to the rest of the pride. In this case, the meme replaces the lion cub with a Shiba Inu dog and the Lion King with a man, possibly Elon Musk, who is holding the dog in his arms. This creates a humorous and unexpected twist on the original scene.

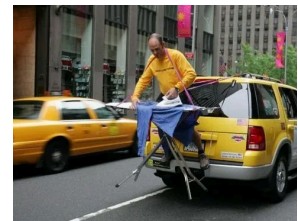

**Query:** What is unusual about this image?

**[Regular]**    GPT-4o-Aided Evaluation: Accuracy:2 Detailedness:3

The unusual aspect of this image is that a man is sitting on a clothes dryer rather than a traditional laundry table. This is considered unusual because clothes dryers are typically found indoors, primarily in laundromats or laundry rooms. The presence of a man sitting on the dryer on a busy street makes this scene even more peculiar and unexpected. It may evoke curiosity about why the person has chosen to do this and what events transpired that lead to this situation.

**[Ours]**    GPT-4o-Aided Evaluation: Accuracy:10 Detailedness:8

The unusual aspect of this image is that a man is ironing clothes while standing on the back of a yellow taxi cab. It is not common to see someone ironing clothes in such a public and unconventional setting, as ironing is typically done indoors, such as in a home or a laundromat. The man's actions might draw attention and curiosity from passersby, as it is not a typical sight in a busy city street.

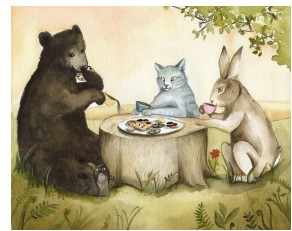

**Query:** What are the animals in the painting and what are they doing?

**[Regular]**    GPT-4o-Aided Evaluation: Accuracy:5 Detailedness:6

The animals in the painting are a white and black cat and a rabbit. They are sitting around a table, eating and drinking from cups and plates.

**[Ours]**    GPT-4o-Aided Evaluation: Accuracy:8 Detailedness:7

The painting features a bear, a cat, and a rabbit. They are all sitting around a table, eating food together.

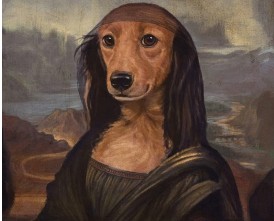

**Query:** What might be the intended effect of this painting?

**[Regular]**    GPT-4o-Aided Evaluation: Accuracy:3 Detailedness:2

The intended effect of this painting could be to introduce the subject, a dog wearing a tie and painting a portrait or causing an artwork resembling a human using oils or a similar medium, portraying itself as the artist. This unusual and creative juxtaposition is intended to provoke curiosity, surprise or amusement among viewers who might not otherwise expect a dog to portray itself as a human artist. Overall, the painting aims to capture viewers' interest and spark conversation or debate about the abstract nature of the scene and the message the artist intended to convey.

**[Ours]**    GPT-4o-Aided Evaluation: Accuracy:9 Detailedness:8

The intended effect of this painting is to create a humorous and unexpected juxtaposition between a human figure and a dog. The dog is depicted wearing a Renaissance-style dress, which is a nod to the famous Mona Lisa painting by Leonardo da Vinci. By placing a dog in the role of the famous painting's subject, the artist challenges the viewer's expectations and creates a lighthearted, comical scene. This painting can evoke amusement and curiosity, as it combines the iconic human figure with a dog, which is not a typical representation in art.

*Figure 9.* **Qualitative LLaVA-Bench comparison (evaluated with GPT-4o).** For each image–query pair, responses from DEGAP compared with those generated by the original LLaVA-1.5. Hallucinated content is highlighted in red, whereas information missed by LLaVA-1.5 but correctly produced by our method appears in blue.

