# OpenReview forum: "DEGAP: Dynamic Entropy-Guided Attention Perturbation for Contrastive Decoding in Large Vision-Language Models"
_ICML.cc/2026/Conference — ICML 2026 regular_

### Official Review · Reviewer_vyNJ · 2026-03-10

**Soundness:** 4
**Presentation:** 4
**Significance:** 4
**Originality:** 3
**Overall Recommendation:** 5
**Confidence:** 4

**Summary:**

The authors propose the Dynamic Entropy-Guided Attention Perturbation (DEGAP) method for contrastive decoding at inference time in LVLMs. The method is motivated by the observation that the divergence between the original and visually masked logits (contrastive logits) starts to emerge when the model’s prediction confidence starts to form. DEGAP then dynamically selects the layer from which all the visual information in the attention is fully masked in the subsequent layers to obtain the contrastive logits. These are then subtracted from the original logits before softmax for the final prediction is applied. Experiments on multiple benchmarks and LVLMs show consistently strong performance.

**Compliance With Llm Reviewing Policy:**

Affirmed.

**Key Questions For Authors:**

- Which k was on average selected? Did you observe any patterns (e.g. certain k for specific data)?
- In Table 4, why do you think that DEGAP favors recall slightly more than precision?
- How does alpha influence the results?

**Limitations:**

/

**Strengths And Weaknesses:**

Strengths:
- Conceptually simple method, showing strong improvements on visual hallucinations.
- Well written motivation in Section 3.3.  The paper is generally easy to follow.
- The dynamic selection for masking layer k is hyperparameter free and transferable to different setups.
- Contrastive forward pass is only partial from the kth layer onwards, saving on the inference latency.


I do not see any major weaknesses. The paper would be stronger with some more ablations studies, for example:
- Impact of full vs partial masking of the visual tokens.
- Impact of alpha.

---

> ### Author Rebuttal · Authors · 2026-03-31
>
> First, we would like to sincerely thank you for the time and effort you devoted to reviewing our work. Your thoughtful and detailed comments address several important aspects of our study, and we greatly appreciate the opportunity to respond to them. Below, we provide our responses to the weaknesses (W) and questions (Q) you raised.
>
> &nbsp;
> # W1
> Thank you for this helpful suggestion. We additionally evaluated different visual-token masking ratios, as shown in Table A.
>
> Among the tested ratios, full masking (100%) achieves the best overall performance, yielding the lowest CHAIR scores and the highest recall. A possible explanation is that partial masking leaves residual visual signals that weaken the contrastive effect, whereas full masking yields a clearer separation between the original and contrast branches. We will include this ablation in the revised version.
>
> | Masking Ratio | $C_S$ ↓ | $C_I$ ↓ | recall ↑ | len |
> |--------------|----------|----------|----------|-----|
> | 20%          | 26.0     | 8.2      | 59.1     | 54.1 |
> | 40%          | 24.4     | 8.2      | 60.9     | 54.4 |
> | 60%          | 23.2     | 7.9      | 62.0     | 54.4 |
> | 80%          | 22.4     | 8.4      | 60.8     | 54.3 |
> | 100% (Ours)  | **19.1**     | **6.3**      | **65.2**     | **54.5** |
>
> Table A. Effect of visual token masking ratio on LLaVA-1.5-7B.
>
> &nbsp;
> # W2 & Q3
>
> Thank you for raising this point. In our method, $\alpha$ controls the strength of the contrastive term in Eq. (9). A larger $\alpha$ places more weight on the contrastive branch, while a smaller $\alpha$ makes decoding closer to regular decoding.
>
> We set $\alpha = 1$ for all tasks, following prior contrastive decoding methods for fair comparison. To address this concern more directly, we additionally conducted a sensitivity analysis on the CHAIR benchmark across multiple $\alpha$ values using LLaVA-1.5-7B.
>
> As shown in Table B, DEGAP consistently outperforms Regular decoding across the tested $\alpha$ values on CHAIR. This suggests that DEGAP remains effective over a reasonably broad range of $\alpha$, rather than relying on a narrowly tuned setting. At the same time, the results indicate that the additional hyperparameter tuning burden is modest in practice. We will include this sensitivity analysis in the revised version.
>
> |$\alpha$|$C_S$ ↓|$C_I$ ↓|
> |:--:|:--:|:--:|
> |0 (Regular)|27.2|9.8|
> |0.25|21|7.4|
> |0.5|21.2|7.5|
> |0.75|20.8|7.5|
> |1.0 (Ours)|**19.1**|**6.3**|
>
> Table B. Sensitivity analysis of $\alpha$.
>
> &nbsp;
> # Q1
>
> Thank you for this important comment. As you pointed out, understanding which perturbatioan start layers are selected in practice is important. We therefore analyzed the distribution of the selected perturbation start layer $l_t^*$ and summarize the layer-wise selection ratios in Table C.
>
> Across all three benchmarks, the selected layer is consistently concentrated in the middle-to-late portion of the decoder, with most selections falling in the 15-24 range. Early layers (0-14) and very deep layers (25 or above) are rarely selected.
>
> We interpret these results as indicating that the entropy-guided layer selection consistently captures the stage where the model's output confidence begins to sharpen.
>
> |Layer Range|CHAIR(%)|POPE(%)|MME(%)|
> |---|---|---|----|
> |≤14|0.33|0.00|0.00|
> |15–19|56.71|56.47|69.61|
> |20–24|41.82|42.68|30.03|
> |≥25|1.15|0.85|0.36|
>
> Table C. Layer selection distribution across benchmarks on LLaVA-1.5-7B.
>
> &nbsp;
> # Q2
>
> Thank you for the insightful question. First, we do not claim that the trend observed in Table 4 is unique to DEGAP. In general, decoding interventions for hallucination mitigation can influence the precision–recall trade-off by suppressing weakly grounded signals. In false-negative cases, the original branch often predicts “no” even when the correct answer is “yes,” especially in borderline situations where the scores for “yes” and “no” are close. DEGAP can be particularly effective in these cases. Although the original branch may capture some positive visual evidence, its final prediction can still lean toward “no” due to a conservative tendency induced by the language prior. Since the masked branch has limited access to visual cues, it tends to reflect this prior-driven tendency more strongly. By contrastively subtracting the masked-branch signal, DEGAP weakens this biased “no” tendency and better preserves visually grounded positive evidence. This can reduce false negatives more effectively, which may explain why, in Table 4, DEGAP shows a relatively larger gain in recall while also improving precision.
>
> &nbsp;
>
> We sincerely thank the reviewer for the thoughtful and constructive feedback. Your comments helped us improve our work and strengthen the revised manuscript. We appreciate the opportunity to refine our study and will address your points clearly in the revision.

---

> > ### Author Rebuttal · Reviewer_vyNJ · 2026-04-07
> >
> > I thank the authors for the extra clarifications. I will keep my original score.

---

> > > ### Author Response · Authors · 2026-04-08
> > >
> > > We sincerely appreciate your time and constructive feedback. We are grateful that our rebuttal was able to address your main concerns, and we will incorporate these clarifications and findings more clearly in the final version.

---

### Official Review · Reviewer_o16x · 2026-03-12

**Soundness:** 3
**Presentation:** 2
**Significance:** 3
**Originality:** 2
**Overall Recommendation:** 4
**Confidence:** 4

**Summary:**

This paper studies hallucination mitigation in large vision-language models via inference-time contrastive decoding. Instead of constructing the contrast branch from externally perturbed images, the authors propose DEGAP, which masks visual attention inside the decoder starting from a dynamically chosen layer determined by a token-wise entropy curve. The paper argues that using layer-wise logit-lens analysis perturbations become effective when entropy starts dropping, and uses this observation to select the masking start layer. Experiments on seven benchmarks and several LVLM backbones show improved hallucination metrics and lower latency than prior contrastive decoding baselines.

**Compliance With Llm Reviewing Policy:**

Affirmed.

**Final Justification:**

The authors have adequately addressed the major technical concerns ann I would like to maintain the score.

**Key Questions For Authors:**

1. In practical inference, how exactly do you compute $l_t^*$ for every token? Are vocabulary projections and entropies computed for all layers at each step？
2. What is the definition of the overlap metric in Figure 4? How are hallucinated objects extracted from the original and contrast branches, and why is overlap with original hallucinations the right target rather than a ground-truth-based criterion?
3. Can you provide a comparison between your entropy-knee rule and simpler alternatives on POPE or MME in addition to CHAIR?

**Limitations:**

Please refer to the above comments.

**Strengths And Weaknesses:**

#### Strength

- The paper tackles a failure mode of hallucinations in LVLMs. The training-free nature of the method and the demonstrated latency reductions offer strong practical utility. The empirical gains on hallucination benchmarks are solid and spread across multiple datasets and backbones.
- Figures 1 and 2 are well-crafted and effectively explain the partial-inference pipeline, making the overall method easy to follow.
- The core idea of building the contrast branch in the attention space rather than via input-image corruption is a well-motivated design choice.

#### Weakness

- The central entropy-guided mechanism remains under-validated. Figure 3 shows correlation rather than necessity or sufficiency, and the paper lacks deeper theoretical or empirical validation to prove that the selected $l_t^*$ is uniquely optimal. The efficiency claim also lacks a detailed breakdown of the per-token layer-wise entropy tracking overhead versus the saved computation.
- Table 3 is confusing, with DEGAP without clear pairwise comparison semantics. Architecture-specific masking details are missing, making it unclear how "visual attention masking" is implemented across diverse backbones whose multimodal fusion mechanisms are not identical.

---

> ### Author Rebuttal · Authors · 2026-03-31
>
> We sincerely thank you for reviewing our work. All of your comments have been helpful, and we will actively incorporate them to further improve the quality of our paper. Below, we respond to the weaknesses (W) and questions (Q) you raised.
>
> &nbsp;
> # W1
> We sincerely thank you for this thoughtful comment. We agree that Figure 3 alone does not prove necessity/sufficiency or the unique optimality of $l_t^*$. Our claim is that entropy serves as a practical, model-agnostic proxy for identifying the prediction formation phase. We will clarify this point and emphasize that our support is empirical, namely the consistent alignment between entropy drop and perturbation sensitivity across models (Figures 3, 5, and 6), as well as the stronger performance of dynamic selection over fixed-layer baselines (Table 6).
>
> We also agree that the efficiency discussion should be clearer. Although entropy tracking introduces a small overhead, our method remains more efficient overall because it reuses the original branch and recomputes only the post-$l_t^*$ layers, instead of requiring an additional full forward pass (Table A). We will revise the paper to clarify this trade-off and highlight our lower latency and memory overhead.
>
> |Method|ms/token|TFLOPs/sample|GPU peak memory(GB)|
> |:---|:---|:---|:---|
> |Regular|53.92|9.72|14.5|
> |VCD|104.41|19.28|15.1|
> |M3ID|104.02|11.20|14.9|
> |DeGF|210.28|29.17|17.7|
> |Ours|88.01|12.96|14.6|
>
> Table A. Efficiency comparison on LLaVA-1.5-7B
>
> &nbsp;
> # W2
> Thank you for this important comment. We agree that Table 3 was not sufficiently clear. It reports one-to-one pairwise comparisons, where DEGAP is evaluated separately against each baseline, rather than a single absolute score. For each pair, GPT-4o compares the two responses and scores them following the official evaluation protocol.
>
> We also clarify that DEGAP does not modify each model’s multimodal fusion module itself. Instead, it is applied uniformly at the decoder level: after the selected layer $l_t^*$, attention scores toward visual tokens are set to -∞, resulting in zero attention weights. Therefore, despite differences in fusion mechanisms across models, the same masking rule can be consistently applied. We will revise the paper to clarify both the Table 3 evaluation protocol and the model-specific application of visual attention masking.
>
> &nbsp;
> # Q1
> As described in Section 3.4, $l_t^*$ is determined at each decoding step by tracking layer-wise logit entropy with the logit lens. Specifically, we project each layer’s hidden state in the original branch to the vocabulary space, compute the entropy, and select the knee point where entropy begins to drop sharply, within the range from the first layer to the minimum-entropy layer.
> Thus, in practical inference, vocabulary projections and entropies are computed for all layers at each decoding step. However, this does not require an additional decoder forward pass, since these computations use the hidden states already produced by the original branch.
>
> &nbsp;
> # Q2
> Thank you for the helpful question. We clarify that the overlap metric in Figure 4 is defined based on object-level hallucinations extracted following the official CHAIR protocol. We extract object mentions from each generated response and regard an object as hallucinated if it is not included in the ground-truth object set.
>
> For Figure 4, we compare hallucinated objects from the regular decoding output (original branch) and from the visually masked output at each masking start layer (contrast branch). The overlap rate is the proportion of hallucinated objects shared by the two branches.
>
> Our goal in Figure 4 is not to evaluate final hallucination performance, but to analyze whether the contrast branch reproduces the hallucination tendency of the original branch. For this reason, overlap with the original hallucinations is the appropriate target: since the contrast branch is intended to expose language-prior-driven errors under weakened visual grounding, higher overlap indicates that it better captures the original branch’s hallucination pattern.
>
> &nbsp;
> # Q3
> We sincerely appreciate this suggestion. To verify that the effectiveness of the entropy-knee rule is not limited to CHAIR, we extended the fixed-layer comparison in Table 6 to POPE and MME (Table B).
>
> The additional results show that fixed-layer masking exhibits substantial performance variation depending on the choice of the start layer, and no fixed layer outperforms the entropy-knee rule. We will include this comparison in the appendix of the revised version.
>
> ||POPE||MME|
> |:--:|:--:|:--:|:--:|
> |Layer|Acc.|F1|MME Score|
> |5|86.71|87.57|584.16|
> |10|87.42|87.77|571.67|
> |15|86.93|87.62|581.94|
> |20|87.56|87.75|596.67|
> |25|87.09|87.85|589.58|
> |Ours|**89.30**|**89.25**|**631.11**|
>
> Table B. Comparison with fixed-layer masking on LLaVA-1.5-7B
>
> &nbsp;
>
> Thank you again for your thoughtful and constructive feedback, which has been invaluable in helping us further develop this work.

---

> > ### Author Rebuttal · Reviewer_o16x · 2026-04-01
> >
> > Thank you for the detailed and thoughtful rebuttal. The clarifications on the role of entropy as a practical proxy, the efficiency trade-offs, and the decoder-level application of the method have addressed my main concerns. The explanation of the evaluation protocol (Table 3) and the additional details on entropy-based layer selection are also helpful.
> >
> > I especially appreciate the additional experiments on POPE and MME. These results provide stronger evidence that the entropy-knee rule generalizes beyond CHAIR. I encourage the authors to include these findings in the main or appendix of the final version to further strengthen the paper.
> >
> > Overall, the rebuttal has improved my confidence in the work, and I will increase my score accordingly.

---

> > > ### Author Response · Authors · 2026-04-02
> > >
> > > Thank you very much for your thoughtful follow-up and for increasing your score. We sincerely appreciate your time and are glad that our rebuttal addressed your main concerns. We will incorporate these clarifications and findings more clearly in the final version.

---

### Official Review · Reviewer_topb · 2026-03-13

**Soundness:** 3
**Presentation:** 2
**Significance:** 2
**Originality:** 3
**Overall Recommendation:** 5
**Confidence:** 4

**Summary:**

They tackle the problem of vision-language model hallucination. Their goal is to detect which next tokens are grounded in the image and which ones are grounded in the previous text. This way, they can reduce the probability of language-only next tokens and reduce hallucinations. To achieve this, they find a layer where the model's entropy (if the subsequent layers until the `lm_head` are skipped) goes down. That layer somewhat shows where the model is starting to make the next token prediction. From that layer onward, they mask all attention to vision tokens and get the next token distribution. They subtract this distribution from the normal output distribution. This subtraction essentially reduces the weights of tokens that are only based on previous language tokens, and not the image.

In their experiments, they show that DEGAP outperforms state-of-the-art baselines across multiple hallucination-specific benchmarks, including CHAIR, POPE, MME, and HallusionBench, and achieves top scores on general-purpose benchmarks like MMBench, MM-Vet, and LLaVA-Bench.

In an important ablation study, they showed that statically choosing the layer to mask attention to the vision model doesn't work well for any single layer.

**Compliance With Llm Reviewing Policy:**

Affirmed.

**Final Justification:**

They addressed all of my concerns.

**Key Questions For Authors:**

1. How did you measure the normal generation time of the model? Did you use vLLM or Hugging Face `.generate()`, for example?

**Limitations:**

No, they didn't. I think having a dedicated section on limitations and, for example, listing the computational overhead would be beneficial.

**Strengths And Weaknesses:**

## Strengths
1. The method is simple, intuitive, and novel.
2. It improves performance considerably.
3. It is a simple, training-free method.
4. The overhead in generation could be minimal in theory since the chosen layer usually happens to be a late layer, and only those layers are being redone.
5. In contrast to previous methods they don't need to augment the input image which is both diffcult and suboptimal

## Weaknesses
1. There is no graph that shows with what distribution different layers are chosen. For example, a graph that shows how many times each layer is chosen.
2. There is around a 2x inference overhead in their implementation.
3. There is no experiment showing the model's sensitivity to the tuning of $\alpha$. If the method's performance is too dependent on that, it could add to hyperparameter difficulty.

---

> ### Author Rebuttal · Authors · 2026-03-31
>
> First, we sincerely thank you for taking the time to review our work. Your insightful comments address several core aspects of our study, and we greatly appreciate the opportunity to discuss them with you. Below, we provide our responses to the weaknesses (W) and questions (Q) you raised.
>
> &nbsp;
>
> # W1
> Thank you for this important comment. As you pointed out, we believe that an analysis of how often each layer is selected is necessary. Accordingly, we summarized the layer-wise selection ratio (%) in Table A.
>
> Across all three datasets, the perturbation start layer ($l_t^*$) is mainly concentrated in the middle-to-late layers, especially in the 15–24 range, while early layers (0–14) and very deep layers (25 or above) are rarely selected.
>
> We interpret these results as indicating that the entropy-guided layer selection stably captures the stage where the model’s output confidence begins to form.
>
> |Layer Range|CHAIR(%)|POPE(%)|MME(%)|
> |---|---|---|----|
> |≤14|0.33|0.00|0.00|
> |15–19|56.71|56.47|69.61|
> |20–24|41.82|42.68|30.03|
> |≥25|1.15|0.85|0.36|
>
> Table A. Layer selection distribution across benchmarks using LLaVA-1.5-7B
>
> &nbsp;
>
> # W2
> As is typical for contrastive decoding-based methods, DEGAP also incurs additional inference cost. However, instead of performing a full additional forward pass to construct the contrast branch, DEGAP is designed to reduce the overhead of a full second pass by reusing the hidden states before the masking start layer and performing partial inference only on the subsequent layers.
>
> To make this point explicit, we additionally measure wall-clock time (ms/token), the average end-to-end TFLOPs per sample, and GPU peak memory usage (GB). As shown in Table B, although DEGAP requires additional computation compared to regular decoding, it achieves the lowest wall-clock latency among the compared contrastive decoding baselines. Moreover, it also exhibits the lowest computational cost in terms of FLOPs, while its GPU peak memory usage remains close to that of regular decoding. Overall, DEGAP maintains a high level of efficiency across latency, computation, and memory usage.
>
> Therefore, while DEGAP does introduce additional computational cost, it provides a more practical efficiency–performance trade-off than existing contrastive decoding-based methods.
>
> |Method|ms/token|TFLOPs/sample|GPU peak memory(GB)|
> |:---:|:---:|:---:|:---:|
> |Regular|53.92|9.72|14.5|
> |VCD|104.41|19.28|15.1|
> |M3ID|104.02|11.20|14.9|
> |DeGF|210.28|29.17|17.7|
> |DEGAP|88.01|12.96|14.6|
>
> Table B. Computational overhead comparison on LLaVA-1.5-7B over the CHAIR
>
> &nbsp;
>
> # W3
> We agree that sensitivity to $\alpha$ is an important practical consideration. In our method, $\alpha$ controls the strength of the contrastive term, and we use $\alpha = 1$ for all tasks following prior contrastive decoding methods for fair comparison. To address this concern more directly, we additionally conducted a sensitivity analysis on CHAIR across multiple $\alpha$ values.
>
> As shown in Table C, DEGAP consistently outperforms Regular decoding on CHAIR across all settings. This suggests that the method remains effective over a reasonably broad range of $\alpha$ values rather than depending on a narrowly tuned setting.
>
> These results indicate that DEGAP is not overly sensitive to the exact choice of $\alpha$, and that the additional hyperparameter tuning burden is small in practical use. We will include this sensitivity analysis in the revised version.
>
> |$\alpha$|$C_S$ ↓|$C_I$ ↓|
> |:--:|:--:|:--:|
> |0 (Regular)|27.2|9.8|
> |0.25|21|7.4|
> |0.5|21.2|7.5|
> |0.75|20.8|7.5|
> |1.0 (Ours)|19.1|6.3|
>
> Table C. Sensitivity analysis of $\alpha$ on LLaVA-1.5-7B
>
> &nbsp;
>
> # Q1
> Thank you for the good question. We did not use vLLM, and all experiments were conducted in the same environment based on Hugging Face’s `.generate()`. Latency was measured using the wall-clock time before and after the `.generate()` call.
>
> &nbsp;
>
> Once again, we sincerely thank you for your careful and insightful feedback. Your comments have provided us with a valuable opportunity to identify and improve the weaker aspects of our work. We will make sure to clearly address the points you raised and appropriately reflect them in the revised manuscript. We truly appreciate the opportunity to engage in this meaningful discussion, and we would be happy to clarify or discuss any additional questions you may have. Thank you again for your time and thoughtful consideration.

---

> > ### Author Rebuttal · Reviewer_topb · 2026-03-31
> >
> > Thanks for your rebuttal. All of my concerns are addressed. I will increase my score as a result.

---

> > > ### Author Response · Authors · 2026-04-01
> > >
> > > Thank you for your thoughtful follow-up. We sincerely appreciate your time and are grateful that our rebuttal was able to address your concerns. Thank you again for your constructive feedback.

---

### Official Review · Reviewer_Jg88 · 2026-03-13

**Soundness:** 2
**Presentation:** 3
**Significance:** 2
**Originality:** 2
**Overall Recommendation:** 3
**Confidence:** 5

**Summary:**

This paper proposes DEGAP, a training-free contrastive decoding method for large vision-language models that constructs the contrast branch by masking visual attention inside the decoder rather than perturbing the input image. The key idea is to dynamically choose the masking start layer at each decoding step using a knee point in the layer-wise entropy curve, then reuse cached hidden states to perform only partial inference for the contrast branch. The paper presents an analysis of layer-wise perturbation effects using logit lens and entropy, and reports results on several benchmarks showing improved hallucination mitigation and competitive or better general VQA performance, with lower latency than prior contrastive decoding methods.

**Compliance With Llm Reviewing Policy:**

Affirmed.

**Final Justification:**

I thank the authors for their effort and clarifications. However, my core concerns remain unresolved. I will maintain my score.

**Key Questions For Authors:**

1. You state that $\alpha$ is set to 1 for all tasks, following VCD. Given that DEGAP perturbs internal representations rather than raw inputs, how sensitive is the model to changes in $\alpha$?
2. At each decoding step $t$, DEGAP fits a two-segment piecewise-linear model for $k \in \{2, \dots, l_{min}-2\}$. What is the exact computational overhead (in FLOPs or wall-clock time) of this specific search step compared to standard decoding?
3. Does the calculated masking start layer $l_{t}^{*}$ vary systematically across different token types (e.g., nouns heavily reliant on visual context versus syntactic function words) during a single generation?
4. How does DEGAP perform on LVLMs with fundamentally different fusion mechanisms?

**Limitations:**

See the weakness

**Strengths And Weaknesses:**

# Strengths
1. The dynamic layer selection algorithm, which fits a two-segment piecewise-linear model to find the knee point, is mathematically principled and avoids brittle, hard-coded thresholds.
2. The manuscript is exceptionally well-structured and clearly written.
3. Linking the model’s predictive confidence (entropy) directly to its reliance on visual grounding to calculate dynamic intervention points is an effective method, although its novelty is limited, given the substantial body of similar existing work, such as DeCo (ICLR 25).

# Weaknesses
1. The paper’s main move is to shift contrastive perturbation from input space to attention space and to choose the intervention layer dynamically. That is a coherent contribution, but the paper sometimes writes as if this is a much larger conceptual leap than it is. Given how many recent decoding-based hallucination papers exist, the burden is on the authors to position the contribution very precisely. The missing related work makes this worse.

2. Equations (5) to (7) define a specific knee-finding rule, and the paper presents it as the key to avoiding brittle fixed-layer choices. But the evidence is limited mainly to Figure 3 and Table 6. Table 6 shows that some fixed layers are worse than dynamic selection, but it does not show whether simpler adaptive heuristics would perform similarly.

3. The paper claims applicability across LLaVA-1.5, InstructBLIP, and mPLUG-OWL2, but the exact masking operation is not described with enough precision for each architecture. “Mask visual attention” sounds simple, but multimodal token integration differs across these models. Without clearer implementation details, reproducibility suffers, and the claimed generality is harder to assess.

4. Lack of advanced base models. The method is only adapted to some outdated structures, such as LLaVA-1.5, InstructBLIP, and mPLUG-OWL2.


5. Figure 3 shows that entropy drop and perturbation sensitivity align, but that does not by itself establish that entropy is the right causal signal for intervention. The paper should tone down language suggesting a stronger principle than the experiments actually establish.

---

> ### Author Rebuttal · Authors · 2026-03-31
>
> We would like to thank you for the effort you devoted to reviewing our work. Below, we provide our responses to the weaknesses(W) and questions(Q) you raised.
>
> &nbsp;
> # W1
> We agree that our method should be situated more precisely within recent research on decoding-time hallucination mitigation. Rather than claiming a fundamentally new paradigm, we view DEGAP as a technical advance in this line of research. Like DeCo(ICLR25) and ANTRP(ICLR25), our method intervenes during decoding using internal model signals. DeCo adaptively integrates information from preceding layers into the final-layer logits, while ANTRP modulates attention eigenspectrum variance related to anchor-token over-propagation. In contrast, DEGAP applies contrastive perturbation directly in attention space and dynamically selects intervention points based on predictive uncertainty. On LLaVA-1.5-7B, DEGAP achieves better results on CHAIR and POPE. We will revise the paper to reflect this more precise positioning.
>
> ||CHAIR||POPE||
> |:--|:--|:--|:--|:--|
> |Method|$C_S$|$C_I$|Acc.|F1|
> |ANTRP|19.5|7.1|84.02|86.42|
> |DeCo|21.4|7.6|87.83|88.19|
> |DEGAP|**19.2**|**6.3**|**88.93**|**89.22**|
>
> &nbsp;
> # W2
> Beyond fixed-layer settings, we also evaluated a simpler adaptive heuristic that selects the first layer whose entropy falls below a preset threshold (CHAIR results on LLaVA-1.5). While this heuristic avoids some poor fixed-layer choices, its performance varies with the threshold and remains below DEGAP overall. Because it depends on absolute entropy values, it requires additional tuning, whereas DEGAP is threshold-free and achieves better CHAIR performance while reducing the brittleness of fixed-layer selection.
>
> |Threshold|$C_S$|$C_I$|
> |:--|:--|:--|
> |8.4|23.0|7.6|
> |8.8|23.4|7.7|
> |9.2|23.4|8.0|
> |9.6|23.0|8.1|
> |DEGAP|**19.1**|**6.3**|
>
> &nbsp;
> # W3
> Although multimodal token integration differs across architectures, DEGAP does not alter the multimodal fusion module itself; instead, it operates only at decoder attention layers, after visual information has been represented as decoder-accessible visual tokens. From the selected layer $l_t^*$ onward, we mask attention from the current query token to visual-token positions by setting the corresponding scores to −∞, while keeping text-text causal attention unchanged.
>
> &nbsp;
> # W4 & Q4
> We agree that evaluation on newer and stronger LVLMs is important. Although fusion mechanisms differ across models, DEGAP operates on decoder-side visual attention after visual token integration and is therefore not tied to a specific upstream fusion design. As a training-free decoding-time method, it can be applied to LVLMs exposing decoder-side visual attention. We additionally evaluated DEGAP on Qwen3-VL-8B and InternVL3.5-8B on CHAIR. DEGAP outperforms all compared baselines. We will include these results in the revised version.
>
> ||Qwen3-VL-8B||InternVL3.5-8B||
> |:--|:--|:--|:--|:--|
> |Method|$C_S$|$C_I$|$C_S$|$C_I$|
> |Regular|12.0|6.9|21.2|8.6|
> |VCD|11.6|6.3|20.8|8.4|
> |M3ID|11.6|6.4|20.4|8.7|
> |DeGF|10.8|6.0|20.1|8.2|
> |DEGAP|**9.2**|**5.2**|**18.6**|**7.1**|
>
> &nbsp;
> # W5
> We agree that the alignment between entropy drop and perturbation sensitivity does not by itself establish entropy as a causal signal for intervention. Instead, we use entropy as a practical proxy for identifying intervention points. This is supported by its consistent alignment with perturbation sensitivity across models (Figures 3, 5, and 6) and by the stronger performance of entropy-guided dynamic selection over static layer selections (Table 6). We will revise the manuscript using more cautious language to reflect this feedback.
>
> &nbsp;
> # Q1
> We set $\alpha=1$ for consistency with prior contrastive decoding work. We additionally evaluated different $\alpha$ values and, as shown in table, DEGAP consistently outperforms Regular decoding on CHAIR across all settings. This suggests that DEGAP isn't overly sensitive to the exact $\alpha$.
>
> |$\alpha$|$C_S$|$C_I$|
> |:--|:--|:--|
> |0(Regular)|27.2|9.8|
> |0.5|21.2|7.5|
> |0.75|20.8|7.5|
> |1.0|19.1|6.3|
>
> &nbsp;
> # Q2
> We separately profiled the knee-finding search over CHAIR and found that the knee-point detection alone took 4.33 ± 0.08 ms/token. Despite this additional step, DEGAP remains more efficient than other contrastive decoding baselines (Table: Overhead comparison on LLaVA-1.5).
>
> |Method|ms/token|TFLOPs/sample|
> |:--|:--|:--|
> |Regular|53.92|9.72|
> |VCD|104.41|19.28|
> |M3ID|104.03|13.20|
> |DeGF|210.29|29.17|
> |DEGAP|88.02|12.96|
>
> &nbsp;
> # Q3
> We analyzed $l_t^*$ on CHAIR generations by comparing object tokens and syntactic function words. We observed a difference: the central 90% of selected layers lies in 19–23 for object tokens and 17–21 for function words. This suggests that entropy-based layer selection adapts to token-level differences in visual dependence.
>
> &nbsp;
>
> Thanks to your suggestion, we further validated our approach and substantially strengthened the revision. We sincerely appreciate the constructive feedback.

---

> > ### Author Rebuttal · Reviewer_Jg88 · 2026-04-04
> >
> > I have significant concerns regarding the novelty of this paper. While DEGAP aims to mitigate LVLM hallucinations via decoding-stage intervention, it shares a striking methodological resemblance to **DeCo (ICLR 2025)**. The complete omission of DeCo—neither cited, analyzed, nor empirically compared—severely undermines the paper's claimed contributions.
> >
> > Specifically, DeCo's core premise is that earlier/middle layers capture correct visual cues that are later suppressed by deep language priors. DeCo addresses this via layer-wise analysis, dynamic layer selection, and decoding correction. DEGAP follows the exact same trajectory: it merely swaps the specific implementation with an entropy-guided knee-point selection, attention masking, and contrastive decoding.
> >
> > Consequently, DEGAP feels less like a novel framework and more like an engineering substitution of the DeCo paradigm. Without acknowledging or comparing against DeCo, it is impossible to evaluate DEGAP’s true incremental value. For a methodology-focused paper, this is a critical flaw.
> >
> > However, the initial submission failed to:
> > * **Acknowledge Prior Work:** Explicitly discuss DeCo and similar paradigms in the related work.
> > * **Clarify Core Differences:** Articulate the fundamental conceptual differences between DEGAP and DeCo, proving this is not just a local variant.
> > * **Empirical Comparison:** Provide direct experimental comparisons and ablation studies against DeCo to demonstrate tangible added value.
> >
> > Additionally, if DEGAP is as highly adaptable as claimed (e.g., ported to Qwen3-VL in days), why were the primary submissions limited to older architectures like LLaVA and InstructBlip? These happen to be the standard testbeds for prior works like DeCo, but evaluating on more recent models like Qwen2.5-VL (or Qwen3-VL, released Sept 2025) would have been far more convincing.
> >
> > Given these major omissions and the lack of proper baselines, this work requires substantial revision and re-evaluation before it can be considered for publication.

---

> > > ### Author Response · Authors · 2026-04-06
> > >
> > > We sincerely thank you for this thoughtful comment.
> > >
> > > We understand why DEGAP and DeCo may appear to belong to the same paradigm, as both involve layer-wise analysis and decoding-time adjustment. However, layer-wise analysis during decoding has also been explored from different perspectives in several hallucination mitigation methods, such as Devils(CVPR 25) and DAMO(ICLR 25). More importantly, DEGAP and DeCo differ in both their starting point and their objective. DeCo focuses on identifying which layer contains the most reliable visual evidence, whereas DEGAP aims to find from which layer visual attention should be perturbed in order to construct a contrast branch that best exposes hallucination.
> > >
> > > Therefore, although both methods examine layers, DEGAP is not simply a local implementation variant of DeCo. Specifically, DeCo is based on the observation that correct visual cues captured in preceding layers become weakened in later layers. It compares token-level distributions across layers, selects a single anchor layer deemed most reliable, and adds its logits to the final logits to reinforce correct tokens. In contrast, DEGAP does not replace one part of DeCo with another. Instead, it automatically determines, based on entropy, the point from which blocking visual information allows the perturbation branch to best reproduce the hallucination tendency of the original distribution. It then contrasts this perturbation branch with the original branch to directly suppress hallucination token logits. In this sense, DeCo is a correction method that reinforces correct tokens, whereas our method is a suppression method that directly reduces hallucination components.
> > >
> > > DeCo selects the anchor layer based on the top candidate tokens from the final layer. As a result, if the correct token has already been pushed out of the candidate set, it is difficult for DeCo to recover it. In addition, because DeCo relies on a single anchor layer, it may not fully exploit cases where the evidence for the correct answer is distributed across multiple layers. In contrast, the layer selection in our method is not a procedure for recovering the information of a particular layer, but rather for dynamically identifying the intervention point at which visual attention masking can form a contrast branch that effectively reflects hallucination tendency. Therefore, DEGAP is better viewed not as a simple variant of DeCo, but as a distinct approach that differs in both the location and mechanism of intervention. Accordingly, in this work, we selected baselines primarily from the contrastive decoding family, which are more directly comparable to our method.
> > >
> > > In fact, our comparison of the layer-selection distributions of DeCo and DEGAP shows that the two methods select clearly different layer ranges (Table A). This suggests that they do not perform the same type of “layer selection,” but instead determine layers for different purposes (i.e., reliable representation selection vs. effective perturbation induction).
> > >
> > > |Selected Layer|DeCo(%)|DEGAP(%)|
> > > |:--|:--|:--|
> > > |≤18|0.0|34.7|
> > > |19|0.0|20.8|
> > > |20|1.8|24.8|
> > > |21|2.6|14.3|
> > > |22|3.8|3.2|
> > > |23|6.1|0.3|
> > > |24|9.6|0.1|
> > > |25|7.8|0.2|
> > > |26|14.7|0.2|
> > > |27|20.7|0.3|
> > > |28|33.0|0.1|
> > > |≥29|0.0|1.1|
> > >
> > > Table A. Layer selection distribution on CHAIR with Qwen3-VL-8B.
> > >
> > > Additionally, we applied DeCo’s layer-selection strategy to DEGAP by constructing the contrast branch with visual attention masking starting from the layer selected by DeCo (Table B). In this case, performance became worse. This further suggests that the layer selection mechanisms of DeCo and DEGAP differ in both purpose and operation, and that DeCo’s criterion is not suitable for DEGAP.
> > >
> > > |Method|$C_S$|$C_I$|
> > > |:--|:--|:--|
> > > |DeCo|9.8|5.5|
> > > |DEGAP|9.2|5.2|
> > > |DeCo+DEGAP|11.2|6.3|
> > >
> > > Table B. DeCo, DEGAP, and DeCo+DEGAP on Qwen3-VL-8B.
> > >
> > > The experiments in the paper were conducted on representative LVLMs widely used in prior work, including DeCo and other prior contrastive decoding methods, in order to evaluate DEGAP under the same settings. At the same time, considering the importance of validation on more recent LVLMs, we had already included results on Qwen2.5-VL (7B, 32B) in Appendix B.4 of the primary submission. Following the reviewer’s suggestion, we agree that evaluation on newer architectures would have further strengthened the submission. We therefore additionally conducted POPE comparisons with baselines on Qwen3-VL, and the results are shown below (Table C). Together with the CHAIR results on Qwen3-VL mentioned in our previous response (Weakness 4), we will include these results in the appendix of the revised version.
> > >
> > > |Method|Acc.|F1|
> > > |:--|:--|:--|
> > > |Regular|88.76|88.21|
> > > |VCD|88.90|88.35|
> > > |M3ID|88.87|88.31|
> > > |DeCo|88.88|89.21|
> > > |DEGAP|89.42|89.29|
> > >
> > > Table C. POPE results on Qwen3-VL-8B.
> > >
> > > Thank you again for your valuable comments, which help improve the completeness of the paper. We hope this response has adequately addressed your concerns about our work.

---

### Decision · Program_Chairs · 2026-04-30

**Decision:**

Accept (regular)

**Comment:**

This paper proposes DEGAP, a contrastive decoding method for hallucination mitigation in LVLMs that constructs the contrast branch by dynamically masking visual attention inside the decoder, with the masking layer selected token-by-token via an entropy-guided knee-point rule. The majority of reviewers recommend acceptance, recognizing the method's practical effectiveness and strong empirical results. The main dissenting concern, raised by reviewer Jg88 and maintained through the full discussion period, is that DEGAP shares substantial methodological overlap with DeCo (ICLR 2025), which was entirely absent from the original submission. This omission was legitimate and flagged independently by another reviewer as well, though that reviewer found the rebuttal's added comparison sufficient to update their assessment. No other reviewer aligned with Jg88's position after cross-review discussion.

I recommend acceptance. The final version must fully incorporate the DeCo comparison and clearly articulate how DEGAP's entropy-guided attention-space perturbation differs mechanistically from DeCo, so that readers can assess the incremental contribution on an informed basis.